# Counterfactual Credit Assignment in Model-Free Reinforcement Learning

## Abstract

Credit assignment in reinforcement learning is the problem of measuring an action's influence on future rewards. In particular, this requires separating *skill* from *luck*, ie. disentangling the effect of an action on rewards from that of external factors and subsequent actions. To achieve this, we adapt the notion of counterfactuals from causality theory to a model-free RL setup. The key idea is to condition value functions on *future* events, by learning to extract relevant information from a trajectory. We then propose to use these as future-conditional baselines and critics in policy gradient algorithms and we develop a valid, practical variant with provably lower variance, while achieving unbiasedness by constraining the hindsight information not to contain information about the agent's actions. We demonstrate the efficacy and validity of our algorithm on a number of illustrative problems.

## 1 Introduction

Reinforcement learning (RL) agents act in their environments and learn to achieve desirable outcomes by maximizing a reward signal. A key difficulty is the problem of *credit assignment* (Minsky, 1961), i.e. to understand the relation between actions and outcomes and to determine to what extent an outcome was caused by external, uncontrollable factors, i.e. to determine the share of 'skill' and 'luck'. One possible solution to this problem is for the agent to build a model of the environment, and use it to obtain a more fine-grained understanding of the effects of an action. While this topic has recently generated a lot of interest (Ha & Schmidhuber, 2018; Hamrick, 2019; Kaiser et al., 2019; Schrittwieser et al., 2019), it remains difficult to model complex, partially observed environments.

In contrast, model-free reinforcement learning algorithms such as policy gradient methods (Williams, 1992; Sutton et al., 2000) perform simple time-based credit assignment, where events and rewards happening after an action are credited to that action, *post hoc ergo propter hoc*. While unbiased in expectation, this coarse-grained credit assignment typically has high variance, and the agent will require a large amount of experience to learn the correct relation between actions and rewards. Another issue of model-free methods is that *counterfactual reasoning*, i.e. reasoning about what would have happened had different actions been taken *with everything else remaining the same*, is not possible. Given a trajectory, model-free methods can in fact only learn about the actions that were actually taken to produce the data, and this limits the ability of the agent to learn quickly. As environments grow in complexity due to partial observability, scale, long time horizons, and large number of agents, actions taken by the agent will only affect a vanishing part of the outcome, making it increasingly difficult to learn from classical reinforcement learning algorithms. We need better credit assignment techniques.

In this paper, we investigate a new method of credit assignment for model-free reinforcement learning which we call *Counterfactual Credit Assignment* (CCA), that leverages hindsight information to implicitly perform counterfactual evaluation - an estimate of the return for actions other than the ones which were chosen. These counterfactual returns can be used to form unbiased and lower variance estimates of the policy gradient by building future-conditional baselines. Unlike classical Q functions, which also provide an estimate of the return for all actions but do so by averaging over all possible futures, our methods provide trajectory-specific counterfactual estimates, i.e. an estimate of the return for different actions, but keeping as many of the external factors constant between the return and its counterfactual estimate. Our method is inspired by ideas from causality theory, but does not require learning a model of the environment. Our main contributions are: a) proposing a set of environments which further our understanding of when difficult credit assignment leads to poor

policy learning; b) introducing new model-free policy gradient algorithms, with sufficient conditions for unbiasedness and guarantees for lower variance. In the appendix, we further c) present a collection of model-based policy gradient algorithms extending previous work on counterfactual policy search; d) connect the literature about causality theory, in particular notions of treatment effects, to concepts from the reinforcement learning literature.

## 2 COUNTERFACTUAL CREDIT ASSIGNMENT

### 2.1 NOTATION

*We use capital letters for random variables and lowercase for the value they take.* Consider a generic MDP $(\mathcal{X}, \mathcal{A}, p, r, \gamma)$. Given a current state $x \in \mathcal{X}$ and assuming an agent takes action $a \in \mathcal{A}$, the agent receives reward $r(x, a)$ and transitions to a state $y \sim p(\cdot|x, a)$. The state (resp. action, reward) of the agent at step $t$ is denoted $X_t$ (resp. $A_t$, $R_t$). The initial state of the agent $X_0$ is a fixed $x_0$. The agent acts according to a policy $\pi$, i.e. action $A_t$ is sampled from the policy $\pi_\theta(\cdot|X_t)$ where $\theta$ are the parameters of the policy, and aims to optimize the expected discounted return $\mathbb{E}[G] = \mathbb{E}[\sum_t \gamma^t R_t]$. The return $G_t$ from step $t$ is $G_t = \sum_{t' \geq t} \gamma^{t'-t} R_{t'}$. Finally, we define the score function $s_\theta(\pi_\theta, a, x) = \nabla_\theta \log \pi_\theta(a|x)$; the score function at time $t$ is denoted $S_t = \nabla_\theta \log \pi_\theta(A_t|X_t)$. In the case of a partially observed environment, we assume the agent receives an observation $E_t$ at every time step, and simply define $X_t$ to be the set of all previous observations, actions and rewards $X_t = (O_{\leq t})$, with $O_t = (E_t, A_{t-1}, R_{t-1})$.[1] $\mathbb{P}(X)$ will denote the probability distribution of a random variable $X$.

### 2.2 POLICY GRADIENT ALGORITHMS

We begin by recalling two forms of policy gradient algorithms and the credit assignment assumptions they make. The first is the REINFORCE algorithm introduced by Williams (1992), which we will also call the single-action policy gradient estimator:

**Proposition 1** (single action estimator). *The gradient of $\mathbb{E}[G]$ is given by $\nabla_\theta \mathbb{E}[G] = \mathbb{E}\left[\sum_{t \geq 0} \gamma^t S_t (G_t - V(X_t))\right]$, where $V(X_t) = \mathbb{E}[G_t|X_t]$.*

The appeal of this estimator lies in its simplicity and generality: to evaluate it, the only requirement is the ability to simulate trajectories, and compute both the score function and the return. Let us note two credit assignment features of the estimator. First, the score function $S_t$ is multiplied not by the whole return $G$, but by the return from time $t$. Intuitively, action $A_t$ can only affect states and rewards coming after time $t$, and it is therefore pointless to credit action $A_t$ with past rewards. Second, removing the value function $V(X_t)$ from the return $G_t$ does not bias the estimator and typically reduces variance. This estimator updates the policy through the score term; note however the learning signal only updates the policy $\pi_\theta(a|X_t)$ at the value taken by action $A_t = a$ (other values are only updated through normalization). The policy gradient theorem from (Sutton et al., 2000), which we will also call all-action policy gradient, shows it is possible to provide learning signal to all actions, given we have access to a Q-function $Q^\pi(x, a) = \mathbb{E}[G_t|X_t = x, A_t = a]$, which we will call a *critic* in the following.

**Proposition 2** (All-action policy gradient estimator). *The gradient of $\mathbb{E}[G]$ is given by $\nabla_\theta \mathbb{E}[G] = \mathbb{E}\left[\sum_t \gamma^t \sum_a \nabla_\theta \pi_\theta(a|X_t) Q^{\pi_\theta}(X_t, a)\right]$.*

A particularity of the all-actions policy gradient estimator is that the term at time $t$ for updating the policy $\nabla \pi_\theta(a|X_t) Q^{\pi_\theta}(X_t, a)$ depends only on past information; this is in contrast with the score function estimates above which depend on the return, a function of the entire trajectory. Proofs can be found in appendix D.1.

### 2.3 INTUITIVE EXAMPLE ON HINDSIGHT REASONING AND SKILL VERSUS LUCK

Imagine a scenario in which Alice just moved to a new city, is learning to play soccer, and goes to the local soccer field to play a friendly game with a group of other kids she has never met. As the game goes on, Alice does not seem to play at her best and makes some mistakes. It turns out however her partner Megan is a strong player, and eventually scores the goal that makes the game a victory. What should Alice learn from this game?

---

[1] Previous actions and rewards are provided as part of the observation as it is generally beneficial to do so in partially observable Markov decision processes.

When using the single-action policy gradient estimate, the outcome of the game being a victory, and assuming a $\pm 1$ reward scheme, all her actions are made more likely; this is in spite of the fact that during this particular game she may not have played well and that the victory is actually due to her strong teammate. From an RL point of view, her actions are wrongly credited for the victory and positively reinforced as a result; effectively, Alice was lucky rather than skillful. Regular baselines do not mitigate this issue, as Alice did not a priori know the skill of Megan, resulting in a guess she had a $50\%$ chance of winning the game and corresponding baseline of $0$. This could be fixed by understanding that Megan's strong play were not a consequence of Alice's play, that her skill was a priori unknown but known in hindsight, and that it is therefore valid to retroactively include her skill level in the baseline. A hindsight baseline, conditioned on Megan's estimated skill level, would therefore be closer to $1$, driving the advantage (and corresponding learning signal) close to $0$.

As pointed out by Buesing et al. (2019), situations in which hindsight information is helpful in understanding a trajectory are frequent. In that work, the authors adopt a model-based framework, where hindsight information is used to ground counterfactual trajectories (i.e. trajectories under *different actions, but same randomness*). Our proposed approach follows a similar intuition, but is model-free: we attempt to *measure*—instead of model— information known in hindsight to compute a *future-conditional baseline*, with the constraint that the captured information must not have been caused by the agent.

### 2.4 FUTURE-CONDITIONAL POLICY GRADIENT ESTIMATOR (FC-PG)

Intuitively, our approach for assigning proper credit to action $A_t$ is as follows: via learning statistics $\Phi_t$ we capture relevant information from the rest of the trajectory, e.g. including observations $O_{t'}$ at times $t'$ greater than $t$. We then learn value functions which are conditioned on the additional hindsight information contained in $\Phi_t$. In general, these future-conditional values and critics would be biased for use in a policy gradient algorithm; we therefore need to correct their impact on the policy gradient through an importance correction term.

**Theorem 1** (Future single-action policy gradient estimator). *Let $\Phi_t$ be an arbitrary random variable. The following is an unbiased estimator of the gradient of $\mathbb{E}[G]$:*

$$\nabla_\theta \mathbb{E}[G] = \mathbb{E}\left[ \sum_t \gamma^t \, S_t \left( G_t - \frac{\pi_\theta(A_t|X_t)}{\mathbb{P}_{\pi_\theta}(A_t|X_t, \Phi_t)} V(X_t, \Phi_t) \right) \right] \quad (1)$$

*where $V(X_t, \Phi_t) = \mathbb{E}[G_t|X_t, \Phi_t]$ is the future $\Phi$-conditional value function[2], and $\mathbb{P}_{\pi_\theta}(A_t|X_t, \Phi_t)$ is the posterior probability of action $A_t$ given $(X_t, \Phi_t)$, for trajectories generated by policy $\pi_\theta$.*

**Theorem 2** (Future all-action policy gradient estimator). *The following is an unbiased estimator of the gradient of $\mathbb{E}[G]$:*

$$\nabla_\theta \mathbb{E}[G] = \mathbb{E}\left[ \sum_t \gamma^t \sum_a \nabla_\theta \log \pi_\theta(a|X_t) \mathbb{P}_{\pi_\theta}(a|X_t, \Phi_t) Q^{\pi_\theta}(X_t, \Phi_t, a) \right] \quad (2)$$

*where $Q^\pi(X_t, \Phi_t, a) = \mathbb{E}[G_t|X_t, \Phi_t, A_t = a]$ is the future-conditional Q function (critic). Furthermore, we have $Q^{\pi_\theta}(X_t, a) = \mathbb{E}\left[ Q^{\pi_\theta}(X_t, \Phi_t, a) \frac{\mathbb{P}_\pi(a|X_t, \Phi_t)}{\pi(a|X_t)} \right]$.*

Proofs can be found in appendix D.2. These estimators bear similarity to (and indeed, generalize) the Hindsight Credit Assignment estimator (Harutyunyan et al., 2019), see the literature review and appendix C for a discussion of the connections.

### 2.5 COUNTERFACTUAL CREDIT ASSIGNMENT POLICY GRADIENT (CCA-PG)

The previous section provides a family of estimators, but does not specify which $\Phi$ should be used, and what type of $\Phi$ would make the estimator useful. Instead of hand-crafting $\Phi$, we will *learn* to extract $\Phi$ from the trajectory (the sequence of observations) $(O_{t'})_{t' \geq 0}$. A useful representation $\Phi$ of the future will simultaneously satisfy two objectives:

- $\Phi_t$ is predictive of the outcome (the return) by learning a $\Phi$-conditional value function, through minimization of $(G_t - V(X_t, \Phi_t))^2$ or $(G_t - Q(X_t, a, \Phi_t))^2$.

---

[2]Note more generally that any function of $X_t$ and $\Phi_t$ can in fact be used as a valid baseline.

- The statistic $\Phi_t$ is 'not a consequence' of action $A_t$; this is done by minimizing (with respect to $\Phi_t$) a surrogate *independence maximization* (IM) loss $\mathcal{L}_{IM}$ which is non-negative and zero if and only if $A_t$ and $\Phi_t$ are conditionally independent given $X_t$.

Intuitively, the statistics $\Phi$ capture exogenous factors to the agent (hence the conditional independence constraint), but that still significantly affect the outcome (hence the return prediction loss). The IM constraint enables us to derive the CCA-PG estimator:

**Theorem 3** (single-action CCA-PG estimator). *If $A_t$ is independent from $\Phi_t$ given $X_t$, the following is an unbiased estimator of the gradient of $\mathbb{E}[G]$:*

$$\nabla_\theta \mathbb{E}[G] = \mathbb{E}\left[\sum_t \gamma^t S_t (G_t - V(X_t, \Phi_t))\right] \tag{3}$$

*Furthermore, the hindsight advantage has no higher variance than the forward one:*
$\mathbb{E}\left[(G_t - V(X_t, \Phi_t))^2\right] \leq \mathbb{E}\left[(G_t - V(X_t))^2\right].$

**Theorem 4** (all-action CCA-PG estimator). *Under the same condition, the following is an unbiased estimator of the gradient of $\mathbb{E}[G]$:*

$$\nabla_\theta \mathbb{E}[G] = \mathbb{E}\left[\sum_t \gamma^t \sum_a \nabla_\theta \pi_\theta(a|X_t) Q^{\pi_\theta}(X_t, \Phi_t, a)\right] \tag{4}$$

*Also, we have for all $a$, $Q^{\pi_\theta}(X_t, a) = \mathbb{E}[Q^{\pi_\theta}(X_t, \Phi_t, a)|X_t, A_t = a].$*

Proofs can be found in appendix D.3. The benefit of the first estimator (equation 3) is clear: under the specified condition, and compared to the regular policy gradient estimator, the CCA estimator is also unbiased, but the variance of its advantage $G_t - V(X_t, \Phi_t)$ (the critical component behind variance of the overall estimator) is no higher.

For the all-action estimator, the benefits of CCA (equation 4) are less self-evident, since this estimator has *higher* variance than the regular all action estimator (which has variance $0$). The interest here lies in bias due to learning imperfect Q functions. Both estimators require learning a Q function from data; any error in Q leads to a bias in $\pi$. Learning $Q(X_t, a)$ requires averaging over all possible trajectories initialized with state $X_t$ and action $a$: in high variance situations, this will require a lot of data. In contrast, if the agent could measure a quantity $\Phi_t$ which has a high impact on the return but is not correlated to the agent action $A_t$, it could be far easier to learn $Q(X_t, \Phi_t, a)$. This is because $Q(X_t, \Phi_t, a)$ computes the averages of the return $G_t$ conditional on $(X_t, \Phi_t, a)$; if $\Phi_t$ has a high impact on $G_t$, the variance of that conditional return will be lower, and learning its average will in turn be simpler. Interestingly, note also that $Q(X_t, \Phi_t, a)$ (in contrast to $Q(X_t, a)$) is a *trajectory-specific* estimate of the return for a counterfactual action.

## 2.6 ALGORITHMIC AND IMPLEMENTATION DETAILS

In this section, we provide one potential implementation of the CCA-PG estimator. Note however than in order to be valid, the estimator only needs to satisfy the conditional independence assumption, and alternative strategies could be investigated. The agent is composed of four components:

- **Agent network**: We assume the agent constructs an internal state $X_t$ from $(O_{t'})_{t' \leq t}$ using an arbitrary network, for instance an RNN, i.e. $X_t = \text{RNN}_\theta(O_t, X_{t-1})$. From $X_t$ the agent computes a policy $\pi_\theta(a|X_t)$.
- **Hindsight network**: Additionally, we assume the agent uses a hindsight network $\varphi$ with parameters which computes a hindsight statistic $\Phi_t = \varphi_\theta((\mathbf{O}, \mathbf{X}, \mathbf{A}))$ (where $(\mathbf{O}, \mathbf{X}, \mathbf{A})$ is the sequence of all observations, agent states and actions in the trajectory), which may depend arbitrarily on the vectors of observations, agent states and actions (in particular, it may depend on observations from timesteps $t' \geq t$). We investigated two architectures. The first is a backward RNN, where $(\Phi_t, B_t) = \text{RNN}_\theta(X_t, B_{t+1})$, where $B_t$ is the state of the backward RNN. Backward RNNs are justified in that they can extract information from arbitrary length sequences, and allow making the statistics $\Phi_t$ a function of the entire trajectory. They also have the inductive bias of focusing more on near-future observations. The second is a transformer (Vaswani et al., 2017; Parisotto et al., 2019). Alternative networks could be used, such as attention-based networks (Hung et al., 2019) or RIMs (Goyal et al., 2019).

- **Value network**: The third component is a future-conditional value network $V_\theta(X_t, \Phi_t)$.

- **Hindsight classifier**: The last component is a probabilistic classifier $h_\omega$ with parameters $\omega$ that takes $X_t, \Phi_t$ as input and outputs a distribution over $A_t$.

Learning is ensured through the minimization of four losses: the hindsight baseline loss $\mathcal{L}_{\text{hs}} = \sum_t (G_t - V_\theta(X_t, \Phi_t))^2$ (optimized with respect to $\theta$); the hindsight classifier loss, $\mathcal{L}_{\text{sup}} = -\sum_t \mathbb{E}[\log h_\omega(A_t|X_t, \Phi_t)]$ (optimized with respect to $\omega$ **only** - all other parameters are treated as constants); the policy gradient surrogate loss $\mathcal{L}_{\text{PG}} = \sum_t \log \pi_\theta(A_t|X_t)\overline{(G_t - V(X_t, \Phi_t))}$, where the bar notation indicates that the quantity is treated as a constant from the point of view of gradient computation; and finally the aforementioned independence loss $\mathcal{L}_{\text{IM}}$, which ensures the conditional independence between $A_t$ and $\Phi_t$. We investigated two IM losses. The first is the Kullback-Leibler divergence between the distributions $\mathbb{P}_{\pi_\theta}(A_t|X_t)$ and $\mathbb{P}_{\pi_\theta}(A_t|X_t, \Phi_t)$. In this case, the KL can be estimated by $\sum_a \mathbb{P}_{\pi_\theta}(a|X_t)(\log \mathbb{P}_{\pi_\theta}(a|X_t) - \log \mathbb{P}_{\pi_\theta}(a|X_t, \Phi_t))$; $\mathbb{P}_{\pi_\theta}(a|X_t)$ is simply the policy $\pi_\theta(a|X_t)$, and the posterior $\mathbb{P}_{\pi_\theta}(a|X_t, \Phi_t)$ can be approximated by probabilistic classifier $h_\omega(A_t|X_t, \Phi_t)$. This results in $\mathcal{L}_{\text{IM}}(t) = \sum_a \pi_\theta(a|X_t)(\log \pi_\theta(a|X_t) - \log h_\omega(a|X_t, \Phi_t))$. We also investigated the conditional mutual information between $A_t$ and $\Phi_t$; again approximated using $h$. We did not see significant differences between the two, with the KL slightly outperforming the mutual information. Finally, note that conversely to the classifier loss, when optimizing the IM loss, $\omega$ is treated as a constant.

Parameter updates and a figure depicting the architecture can be found in Appendix A.

## 3 NUMERICAL EXPERIMENTS

Given its guarantees on lower variance and unbiasedness, we run all our experiments on the single action version of CCA-PG.

### 3.1 BANDIT WITH FEEDBACK

We first demonstrate the benefits of hindsight value functions in a toy problem designed to highlight these. We consider a contextual bandit problem with feedback. Given $N, K \in \mathbb{N}$, we sample for each episode an integer context $-N \leq C \leq N$ as well as an exogenous noise $\epsilon_r \sim \mathcal{N}(0, \sigma_r)$. Upon taking action $A \in \{-N, \dots, N\}$, the agent receives a reward $R = -(C-A)^2 + \epsilon_r$. Additionally, the agent is provided with a $K$-dimensional feedback vector $F = U_C + V_A + W\epsilon_r$ where $U_n, V_n \in \mathbb{R}^K$ for $-N \leq n \leq N$, and $W \in \mathbb{R}^K$ are fixed vectors; in our case, for each seed, they are sampled from standard Gaussian distribution and kept constant through all episodes. More details about this problem as well as variants are presented in Appendix B.1.

For this problem, the optimal policy is to choose $A = C$, resulting in average reward of $0$. However, the reward $R$ is the sum of the informative reward $-(C-A)^2$ and the noisy reward $\epsilon_r$, uncorrelated to the action. The higher the standard deviation $\sigma_r$, the more difficult it is to perform proper credit assignment, as high rewards are more likely due to a high value of $\epsilon_r$ than an appropriate choice of action. On the other hand, the feedback $F$ contains information about $C$, $A$ and $\epsilon_r$. If the agent can extract information $\Phi$ from $F$ in order to capture information about $\epsilon_r$ and use it to compute a hindsight value function, the effect of the perturbation $\epsilon_r$ may be removed from the advantage, resulting in a significantly lower variance estimator. However, if the agent blindly uses $F$ to compute the hindsight value information, information about the context and action will 'leak' into the hindsight value, leading to an advantage of $0$ and no learning: intuitively, the agent will assume the outcome is entirely controlled by chance, and that all actions are equivalent, resulting in a form of learned helplessness.

We investigate the proposed algorithm with $N = 10, K = 64$. As can be seen on Fig. 1, increasing the variance of the exogenous noise leads to dramatic decrease of performance for the vanilla PG estimator without the hindsight baseline; in contrast, the CCA-PG estimator is generally unaffected by the exogenous noise. For very low level of exogenous noise however, CCA-PG suffers from a decrease in performance. This is due to the agent computing a hindsight statistic $\Phi$ which is not perfectly independent from $A$, leading to bias in the policy gradient update. The agent attributes part of the reward to chance, despite the fact that in low-noise regime, the outcome is entirely due to the agent's action. To demonstrate this, and evaluate the impact of the independence constraint on performance, we run CCA-PG with different values of the weight $\lambda_{\text{IM}}$ of the independence max-

imization loss, as seen in Fig. 1. For lower values of this parameter, i.e. when $\Phi$ and $A$ have a larger mutual information, the performance is dramatically degraded.

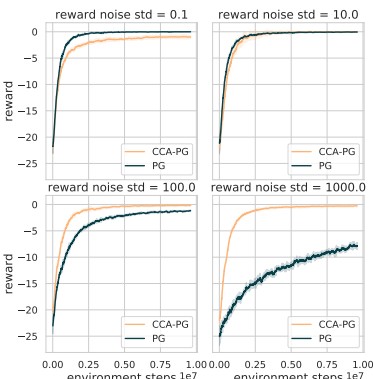 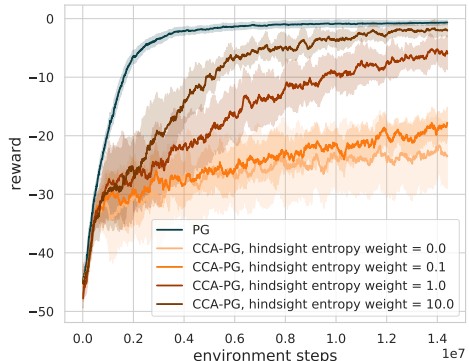

**Figure 1: Left:** Comparison of CCA-PG and PG in contextual bandits with feedback, for various levels of reward noise $\sigma_r$. Results are averaged over 6 independent runs. **Right:** Performance of CCA-PG on the bandit task, for different values of $\lambda_{\text{IM}}$. Not properly enforcing the independence constraint results in strong degradation of performance.

## 3.2 KEY-TO-DOOR ENVIRONMENTS

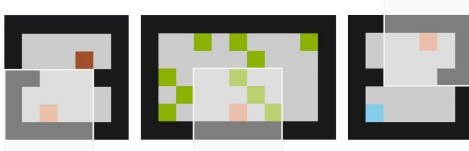

**Figure 2: Key-To-Door environments visual.** The agent is represented by the beige pixel, key by brown, apples by green, and the final door by blue. The agent has a partial field of view, highlighted in white.

**Task Description.** We introduce the Key-To-Door family of environments as a testbed of tasks where credit assignment is hard and is necessary for success. In this environment (cf. Fig. 2), the agent has to pick up a key in the first room, for which it has *no immediate reward*. In the second room, the agent can pick up 10 apples, that each give *immediate rewards*. In the final room, the agent can open a door (only if it has picked up the key in the first room), and receive a small reward. In this task, a single action (i.e picking up the key) has a very small impact on the reward it receives in the final room, while its episode return is largely driven by its performance in the second room (i.e picking up apples).

We now consider two instances of the Key-To-Door family that illustrate the difficulty of credit assignment in the presence of extrinsic variance. In the Low-Variance-Key-To-Door environment, each apple is worth a reward of 1 and opening the final door also gets a reward of 1. Thus, an agent that solves the apple phase perfectly sees very little variance in its episode return and the learning signal for picking up the key and opening the door is relatively strong.

High-Variance-Key-To-Door keeps the overall structure of the Key-To-Door task, but now the reward for each apple is randomly sampled to be either 1 or 10, and fixed within the episode. In this setting, even an agent that has a perfect apple-phase policy sees a large variance in episode returns, and thus the learning signal for picking up the key and opening the door is comparatively weaker. Appendix B.2.1 has some additional discussion illustrating the difficulty of learning in such a setting.

**Results** We test CCA-PG on our environments, and compare it against Actor-Critic (Williams (1992), as well as State-conditional HCA and Return-conditional HCA (Harutyunyan et al., 2019) as baselines. We test using both a backward-LSTM (referred to as CCA-PG RNN) or an attention model (referred to as CCA-PG Attn) for the hindsight function. Details for experimental setup are provided in Appendix B.2.2. All results are reported as median performances over 10 seeds.

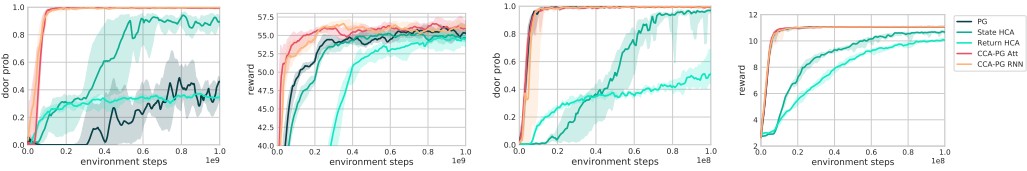

**Figure 3:** Probability of opening the door and total reward obtained on the **High-Variance-Key-To-Door** task (left two) and the **Low-Variance-Key-To-Door** task (right two).

We evaluate agents both on their ability to maximize total reward, as well as solve the specific credit assignment problem of picking up the key and opening the door. Figure 3 compares CCA-PG with the baselines on the High-Variance-Key-To-Door task. Both CCA-PG architectures outperform the baselines in terms of total reward, as well as probability of picking up the key and opening the door.

This example highlights the capacity of CCA-PG to learn and incorporate trajectory-specific external factors into its baseline, resulting in lower variance estimators. Despite being a difficult task for credit assignment, CCA-PG is capable of solving it quickly and consistently. On the other hand, vanilla actor-critic is greatly impacted by this external variance, and needs around $3.10^9$ environment steps to have an 80% probability of opening the door. CCA-PG also outperforms State- and Return-Conditional HCA, which do use hindsight information but in a more limited way than CCA-PG.

On the Low-Variance-Key-To-Door task, due to the lack of extrinsic variance, standard actor-critic is able to perfectly solve the environment. However, it is interesting to note that CCA-PG still matches this perfect performance. On the other hand, the other hindsight methods struggle with both door-opening and apple-gathering. This might be explained by the fact that both these techniques do not guarantee lower variance, and rely strongly on their learned hindsight classifiers for their policy gradient estimators, which can be harmful when these quantities are not perfectly learned. See Appendix B.2.3 for additional experiments and ablations on these environments.

These experiments demonstrate that CCA-PG is capable of efficiently leveraging hindsight information to mitigate the challenge of external variance and learn strong policies that outperform baselines. At the same time, it suffers no drop in performance when used in cases where external variance is minimal.

### 3.3 TASK INTERLEAVING

**Motivation.** In the real world, human activity can be seen as solving a large number of loosely related problems. These problems are not solved sequentially, as one may temporarily engage with a problem and only continue engaging with it or receive feedback from its earlier actions significantly later. At an abstract level, one could see this lifelong learning process as solving problems not in a sequential, but an interleaved fashion instead. The structure of this interleaving also will typically vary over time. Despite this very complex structure and receiving high variance rewards from the future, humans are able to quickly make sense of these varying episodes and correctly credit their actions. This learning paradigm is quite different from what is usually considered in reinforcement learning. Indeed, focus is mostly put on agents trained on a single task, with an outcome dominated by the agent's actions, where long term credit assignment is not required and where every episode will be structurally the same. To the end of understanding the effects of this interleaving on lifelong learning, we introduce a new class of environments capturing the structural properties mentioned above. In contrast to most work on multi-task learning, we do not assume a clear delineation between subtasks —each agent will encounter multiple tasks in a single episode, and it is the agent's responsibility to implicitly detect boundaries between them.

**Task Description.** As described in Fig. 4, this task consists of interleaved pairs of query-answer rooms with different visual contexts that represents different tasks. Each task has an associated mapping of 'good' (resp. 'bad') colors yielding to high (resp. zero) reward. Each episode is composed of *randomly sampled tasks* and color pairs within those tasks. The ordering and the composition of each episode is random across tasks and color pairs. A visual example of what an episode looks like can be seen in Fig. 4. Additional details are provided in B.3.1.

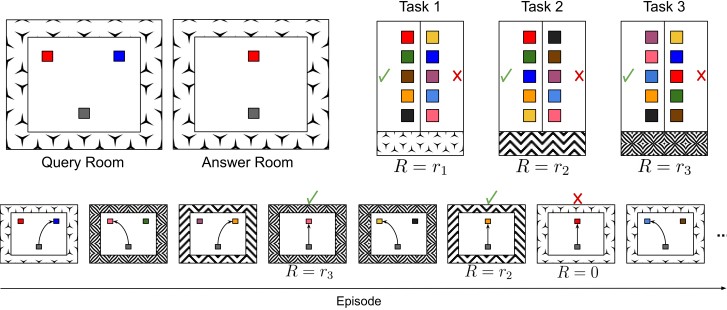

**Figure 4: Multi Task Interleaving Description.** Top left. Delayed feedback contextual bandit problem. Given a context shown as a surrounding visual pattern, the agent has to decide to pick up one of the two colored squares where only one will be rewarding. The agent is later teleported to the second room where it is provided with its previous choice and a visual cue about which colored square it should have picked up. Top right: Different tasks which each a different color mapping, visual context and associated reward. Bottom: Example of a generated episode, composed of randomly sampled tasks and color pairs.

The 6 tasks we will consider next (numbered #1 to #6) are respectively associated with a reward of 80, 4, 100, 6, 2 and 10. Tasks #2, #4, #5 and #6 are referred to as 'hard' while tasks #1 and #3 as 'easy' because of their large associated rewards. The settings 2, 4 and 6-task are respectively considering tasks 1-2, 1-4 and 1-6. In addition to the total reward, we record the probability of picking up the correct square for the easy and hard tasks separately. Performance in the hard tasks will indicate ability to do fine-grained credit assignment.

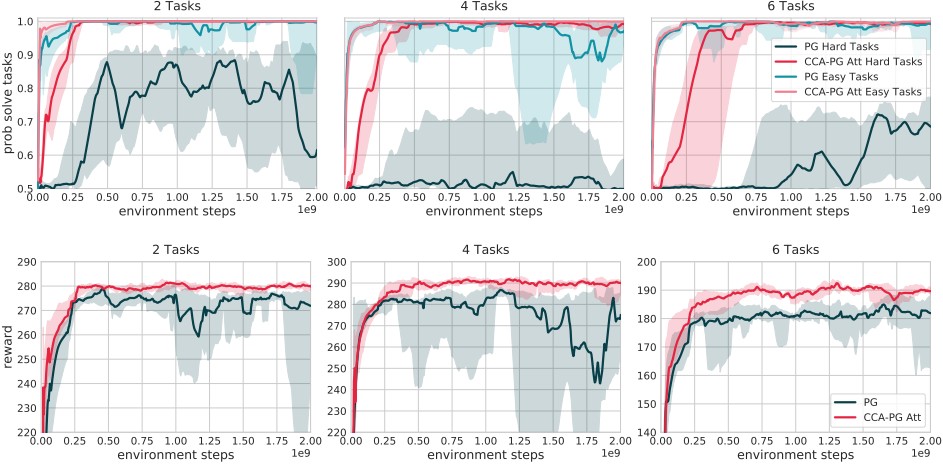

**Figure 5:** Probability of solving 'easy' and 'hard' tasks and total reward obtained for the **Multi Task Interleaving.** Top plot: Median over 10 seeds after doing a mean over the performances in 'easy' or 'hard' tasks.

**Results.** While CCA-PG is able to perfectly solve both the 'easy' and 'hard' tasks in the three setups in less than $5.10^8$ environment steps (Fig. 5), actor-critic is only capable to solve the 'easy' tasks for which the associated rewards are large. Even after $2.10^9$ environment steps, actor-critic is still greatly impacted by the variance and remains incapable of solving 'hard' tasks in any of the three settings. CCA also outperforms actor-critic in terms of the total reward obtained in each setting. State-conditional and Return-conditional HCA were also evaluated on this task but results are not reported as almost no learning was taking place on the 'hard' tasks. Details for experimental setup are provided in B.3.2. All results are reported as median performances over 10 seeds. More results along with an ablation study can be found in B.3.3.

Through efficient use of hindsight, CCA-PG is able to take into account trajectory-specific factors such as the kinds of rooms encountered in the episode and their associated rewards.

In the case of the Multi-Task Interleaving environment, an informative hindsight function would capture the reward for different contexts and exposes as $\Phi_t$ all rewards obtained in the episode except those associated with the current context. This experiment again highlights the capacity of CCA-PG to solve hard credit assignment problems in a context where the return is affected by multiple distractors, while PG remains highly sensitive to them.

## 4    RELATED WORK

This paper builds on work from Buesing et al. (2019) which shows how causal models and real data can be combined to generate counterfactual trajectories and perform off-policy evaluation for RL. Their results however require an explicit model of the environment. In contrast, our work proposes a model-free approach, and focuses on policy improvement. Oberst & Sontag (2019) also investigate counterfactuals in reinforcement learning, point out the issue of non-identifiability of the correct SCM, and suggest a sufficient condition for identifiability; we discuss this issue in appendix F. Closely related to our work is Hindsight Credit Assignment, a concurrent approach from Harutyunyan et al. (2019); in this paper, the authors also investigate value functions and critics that depend on future information. However, the information the estimators depend on is hand-crafted (future state or return) instead of arbitrary functions of the trajectory; their estimators is not guaranteed to have lower variance. Our FC estimator generalizes their estimator, and CCA further characterizes which statistics of the future provide a useful estimator. Relations between HCA, CCA and FC are discussed in appendix C. The HCA approach is further extended by Young (2019), and Zhang et al. (2019) who minimize a surrogate for the variance of the estimator, but that surrogate cannot be guaranteed to actually lower the variance. Similarly to state-HCA, it treats each reward separately instead of taking a trajectory-centric view as CCA. Guez et al. (2019) also investigate future-conditional value functions; similar to us, they learn statistics of the future $\Phi$ from which returns can be accurately predicted, and show that doing so leads to learning better representations (but use regular policy gradient estimators otherwise). Instead of enforcing a information-theoretic constraint, they bottleneck information through the size of the encoding $\Phi$. In domain adaptation (Ganin et al., 2016; Tzeng et al., 2017), robustness to the training domain can be achieved by constraining the agent representation not to be able to discriminate between source and target domains, a mechanism similar to the one constraining hindsight features not being able to discriminate the agent's actions.

Both Andrychowicz et al. (2017) and Rauber et al. (2017) leverage the idea of using hindsight information to learn goal-conditioned policies. Hung et al. (2019) leverage attention-based systems and episode memory to perform long term credit assignment; however, their estimator will in general be biased. Ferret et al. (2019) looks at the question of transfer learning in RL and leverage transformers to derive a heuristic to perform reward shaping. Arjona-Medina et al. (2019) also addresses the problem of long-term credit assignment by redistributing delayed rewards earlier in the episode; their approach still fundamentally uses time as a proxy for credit.

Previous research also leverages the fact that baselines can include information unknown to the agent at time $t$ (but potentially revealed in hindsight) but not affected by action $A_t$, see e.g. (Wu et al., 2018; Foerster et al., 2018; Andrychowicz et al., 2020; Vinyals et al., 2019). Note however that all of these require privileged information, both in the form of feeding information to the baseline inaccessible to the agent, and in knowing that this information is independent from the agent's action $A_t$ and therefore won't bias the baseline. Our approach seeks to replicate a similar effect, but in a more general fashion and from an agent-centric point of view, where the agent *learns itself* which information from the future can be used to augment its baseline at time $t$.

## 5    CONCLUSION

In this paper we have considered the problem of credit assignment in RL. Building on insights from causality theory and structural causal models we have developed the concept of future-conditional value functions. Contrary to common practice these allow baselines and critics to condition on future events thus separating the influence of an agent's actions on future rewards from the effects of other random events thus reducing the variance of policy gradient estimates. A key difficulty lies in the fact that unbiasedness relies on accurate estimation and minimization of mutual information. Learning inaccurate hindsight classifiers will result in miscalibrated estimation of luck, leading to bias in learning. Future research will investigate how to scale these algorithms to more complex environments, and the benefits of the more general FC-PG and all-actions estimators.

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

# Appendix

## A ARCHITECTURE

The parameter updates are as follows:

**Parameter updates**    For each trajectory $(X_t, A_t, R_t)_{t \geq 0}$, compute the parameter updates :

- $\Delta\theta = -\lambda_{\text{PG}} \sum_t \nabla_\theta \log \pi_\theta(A_t | X_t)(G_t - V(X_t, \Phi_t)) - \lambda_{\text{hs}} \nabla_\theta \mathcal{L}_{\text{hs}}(t) - \lambda_{\text{IM}} \nabla_\theta \sum_t \mathcal{L}_{\text{IM}}(t)$
- $\Delta\omega = -\nabla_\omega \mathcal{L}_{\text{sup}}(t)$

where the different $\lambda$ are the weights of each loss.

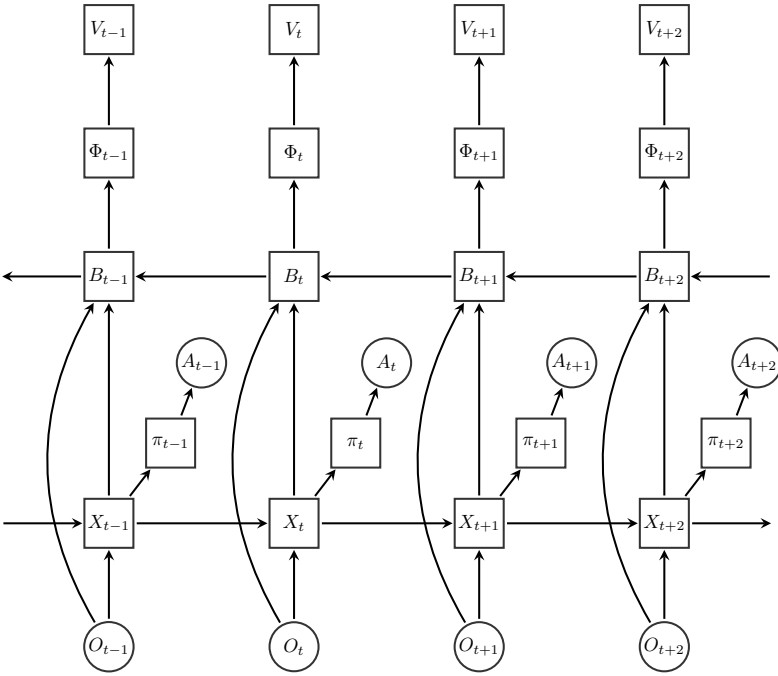

**Figure 6:** Overall architecture for the RNN network. For simplicity we assume without loss of generality that $B_t$ and $\Phi_t$ include $X_t$.

## B ADDITIONAL EXPERIMENTAL DETAILS

### B.1 BANDITS

#### B.1.1 ARCHITECTURE

For the bandit problems, the agent architecture is as follows:

- The hindsight feature $\Phi$ is computed by a backward RNN. We tried multiple cores for the RNN: GRU ( (Chung et al., 2015) with 32 hidden units, a recurrent adder ($b_t = b_{t-1} + \text{MLP}(x_t)$, where the MLP has two layers of 32 units), or an exponential averager ($b_t = \lambda b_{t-1} + (1 - \lambda)\text{MLP}(x_t)$).
- The hindsight classifier $h_\omega$ is a simple MLP with two hidden layers with 32 units each.
- The policy and value functions are computed as the output of a simple linear layer with concatenated observation and feedback as input.
- All weights are jointly trained with Adam (Kingma & Ba, 2014).
- Hyperparameters are chosen as follows (unless specified otherwise): learning rate $4e\text{-}4$, entropy loss $4e\text{-}3$, independence maximization tolerance $\beta_{\text{IM}} = 0.1$; $\lambda_{\text{fwd}} = \lambda_{\text{hw}} = 1$; $\lambda_{\text{IM}}$ is set through Lagrangian optimization (GECO, Rezende & Viola (2018)).

### B.1.2 ADDITIONAL RESULTS

**Multiagent Bandit Problem:**    In the multi-agent version, which we will call MULTI-BANDIT, the environment is composed of $M$ replicas of the bandit with feedback task. Each agent $i = 1, \ldots, M$ interacts with its own version of the environment, but feedback and rewards are coupled across agents; MULTI-BANDIT is obtained by modifying the single agent version as follows:

- The contexts $C^i$ are sampled i.i.d. from $\{-N, \ldots, N\}$. $C$ and $A$ now denote the concatenation of all agents' contexts and actions.
- The feedback tensor is $(M, K)$ dimensional, and is computed as $W_c \mathbf{1}(C) + W_a \mathbf{1}(A) + \epsilon_f$; where the $W$ are now three dimensional tensors. Effectively, the feedback for agent $i$ depends on the context and actions of all other agents.
- The observation for agent $i$ at step $t \geq 1$ is $(0, F[t])$, where $F[t] = (F_{i,(t-1)B+1:tB})$.
- The terminal joint reward is $\sum_i -(C^i - A_0^i)^2$ for all agents.

The multi-agent version does not require the exogenous noise $\epsilon_e$, as other agents play the role of exogenous noise; it is a minimal implementation of the example found in section 2.3.

Finally, we report results from the MULTI-BANDIT version of the environment, which can be found in Fig. 7. As the number of interacting agents increases, the effective variance of the vanilla PG estimator increases as well, and the performance of each agent decreases. In contrast, CCA-PG agents learn faster and reach higher performance (though they never learn the optimal policy).

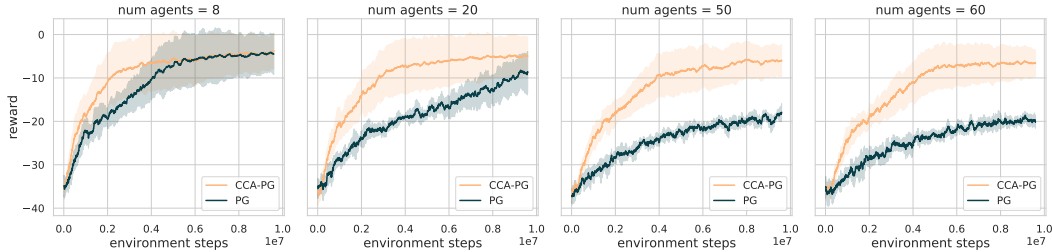

**Figure 7: Multiagent versions of the bandit problems.** CCA-PG agents outperform PG in the single timestep version.

## B.2   KEY TO DOOR TASKS

### B.2.1   ENVIRONMENT DETAILS

Table 1 shows the advantages for either picking up the key or not, for an agent that has a perfect apple-phase policy, but never picks up the key or door, on High-Variance-Key-To-Door. Since there are 10 apples which can be worth 1 or 10, the return will be either 10 or 100. Thus the forward baseline in they key phase, i.e. before it has seen how much an apple is worth in the current episode, will be 55. As seen here, the difference in advantages due to Luck is far larger than the difference in advantage due to Skill when not using hindsight, making learning difficult, leading to the policy never learning to start picking up the key or door. However, when we use a hindsight-conditioned baseline, we are able to learn a $\Phi$ (i.e. the value of a single apple in the current episode) that is completely independent from the actions taken by the agent, but which can provide a perfect hindsight-conditioned baseline of either 10 or 100.

|  |  | Lucky (high apple reward) | Unlucky (low apple reward) |
|---|---|---|---|
| Hindsight advantage | Skillful (Got key + Door) | 1 | 1 |
|  | Unskillful (Did not get key or door) | 0 | 0 |
| Forward Advantage | Skillful (Got key + Door) | 46 | -44 |
|  | Unskillful (Did not get key or door) | 45 | -45 |

**Table 1:** The advantage of the action of picking up a key in High-Variance-Key-To-Door, as computed by an agent that always picks up every apple, and never picks up the key or door. We see that an advantage learned using hindsight clearly differentiates between the skillful and unskillful actions; whereas for an advantage learned without using hindsight, this difference is dominated by the extrinsic randomness.

### B.2.2 ARCHITECTURE

The agent architecture is as follows:

- The observation are first fed to 2-layer CNN with with $(16, 32)$ output channels, kernel shapes of $(3, 3)$ and strides of $(1, 1)$. The output of the CNN is the flattened and fed to a linear layer of size $128$.

- The agent state is computed as a forward LSTM with a state size of $128$. The input to the LSTM are the output of the previous linear layer, concatenated with the reward at the previous timestep.

- The hindsight feature $\Phi$ is computed either by a backward LSTM (i.e CCA-PG RNN) with a state size of $128$ or by an attention mechanism Vaswani et al. (2017) (i.e CCA-PG Att) with value and key sizes of $64$, 1 transformer block with 2 attention heads and a 1 hidden layer mlp of size $1024$, an output size of $128$ and a rate of dropout of $0.1$. The input provided is the concatenation of the output of the forward LSTM and the reward at the previous timestep.

- The policy is computed as the output of a simple MLP with one layer with $64$ units where the output of the forward LSTM is provided as input.

- The forward baseline is computed as the output of a 3-layer MLP of $128$ units each where the output of the forward LSTM is provided as input.

- For CCA, the hindsight classifier $h_\omega$ is computed as concatenation of the output of an MLP, with four hidden layers with $256$ units each where the the concatenation of the output of the forward LSTM and the hindsight feature $\Phi$ is provided as input, and the log of the policy outputs.

- For State HCA, the hindsight classifier $h_\omega$ is computed as the output of an MLP, with four hidden layers with $256$ units each where the the concatenation of the outputs of the forward LSTM at two given time steps is provided as input.

- For Return HCA, the hindsight classifier $h_\omega$ is computed as the output of an MLP, with four hidden layers with $256$ units each where the the concatenation of the output of the forward LSTM and the return is provided as input.

- The hindsight baseline is computed as the output of a 3-layer MLP of $128$ units each where the concatenation of the output of the forward LSTM and the hindsight feature $\Phi$ is provided as input. The hindsight baseline is trained to learn the residual between the return and the forward baseline.

- All weights are jointly trained with RMSprop (Hinton et al., 2012) with epsilon $1e\text{-}4$, momentum 0 and decay $0.99$.

|                          | CCA Att | CCA LSTM | PG   | State HCA | Return HCA |
|--------------------------|---------|----------|------|-----------|------------|
| Policy cost              | 1       | 1        | 1    | 1         | 1          |
| Entropy Cost             | 5e-3    | 5e-3     | 5e-3 | 5e-3      | 5e-3       |
| Forward baseline cost    | 5e-2    | 5e-2     | 5e-2 | 5e-2      | 5e-2       |
| Conditional baseline cost| 5e-2    | 5e-2     | —    | ——        | ——         |
| Hindsight classifier cost| 1e-2    | 1e-2     | —    | 1e-2      | 1e-2       |
| Action independence cost | 1e2     | 1e2      | —    | ——        | ——         |
| Learning rate            | 5e-4    | 5e-4     | 1e-4 | 5e-4      | 1e-3       |

**Table 2:** High-Variance-Key-To-Door hyperparameters

|                          | CCA Att | CCA LSTM | PG   | State HCA | Return HCA |
|--------------------------|---------|----------|------|-----------|------------|
| Policy cost              | 1       | 1        | 1    | 1         | 1          |
| Entropy Cost             | 5e-3    | 5e-3     | 5e-3 | 5e-3      | 5e-3       |
| Forward baseline cost    | 5e-2    | 5e-2     | 5e-2 | 5e-2      | 5e-2       |
| Conditional baseline cost| 5e-2    | 5e-2     | —    | ——        | ——         |
| Hindsight classifier cost| 1e-2    | 1e-2     | —    | 1e-2      | 1e-2       |
| Action independence cost | 1e2     | 1e2      | —    | ——        | ——         |
| Learning rate            | 5e-4    | 5e-4     | 5e-4 | 5e-4      | 5e-4       |

**Table 3:** Key-To-Door hyperparameters

For High-Variance-Key-To-Door, the optimal hyperparameters found and used for each algorithm can be found in Table 2.

For Key-To-Door, the optimal hyperparameters found and used for each algorithm can be found in Table 3.

The agents are trained on full-episode trajectories, using a discount factor of 0.99.

### B.2.3  ADDITIONAL RESULTS

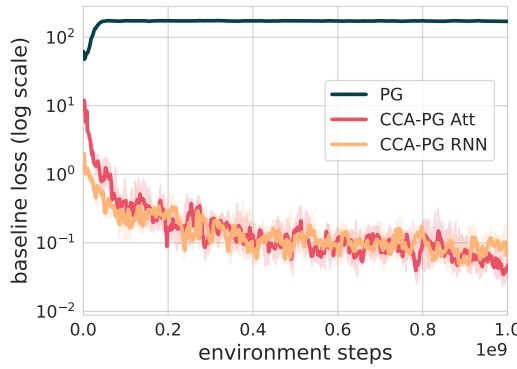

**Figure 8:** Baseline loss for policy gradient versus conditioned baseline loss for CCA in **High Variance Key To Door.**

As shown in Fig. 8, in the case of of actor-critic, the baseline loss increases at first. As the reward associated with apples vary from one episode to another, getting more apples also means increasing the forward baseline loss. On the other hand, as CCA is able to take into account trajectory specific exogenous factors, the hindsight baseline loss can nicely decrease as learning takes place.

Fig. 9 shows the impact of the variance level induced by the apple reward discrepancy between episodes on the probability of picking up the key and opening the door. Thanks to the use of hindsight in its value function, CCA-PG is almost not impacted by this whereas actor-critic sees its performances drops dramatically as variance increases.

Figure 10 shows a qualitative analysis of the attention weights learned by CCA-PG Att on the High-Variance-Key-To-Door task. For this experiment, we use only a single attention head for easier interpretation of the hindsight function, and show both a heatmap of the attention weights over the entire episode, and a histogram of attention weights at the step where the agent picks up the key. As

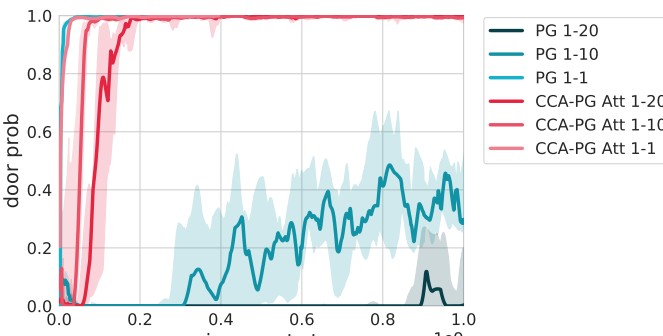

**Figure 9: Impact of variance over credit assignment performances.** Probability of opening the door and total reward obtained as a function of the variance level induced by the apple reward discrepancy.

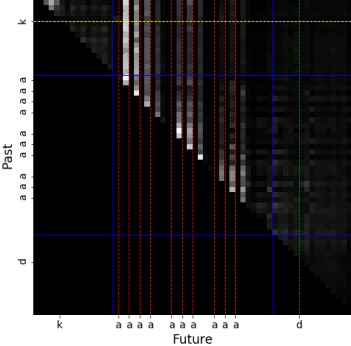

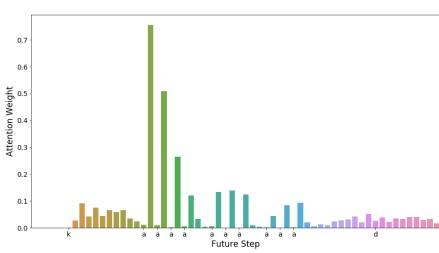

**Figure 10:** Visualization of attention weights on the High-Variance-Key-To-Door task. **Left:** a 2-dimensional heatmap showing how the hindsight function at each step attends to each step in the future. Red lines indicate the timesteps at which apples are picked up (marked as 'a'); green indicates the door (marked as 'd'); yellow indicates the key (marked as 'k'). **Right:** A bar plot of attention over future timesteps, computed at the step where the agent is just about to pick up the key.

expected, the most attention is paid to timesteps just after the agent picks up an apple - since these are the points at which the apple reward is provided to the $\Phi$ computation. In particular, very little attention is paid to the timestep where the agent opens the door. These insights further show that the hindsight function learned is highly predictive of the episode return, while not having mutual information with the action taken by the agent, thus ensuring an unbiased policy gradient estimator.

### B.3 MULTI TASKS INTERLEAVING

#### B.3.1 ENVIRONMENT DETAILS

For each task, a random set, but fixed through training, set of 5 out of 10 colored squares are leading to a positive reward. Furthermore, a small reward of 0.5 is provided to the agent when it picks up any colored square. Each episode are 140 steps long and it takes 9 steps for the agent to reach one colored square from it initial position.

#### B.3.2 ARCHITECTURE

We use the same architecture setup as reported in Appendix B.2.2. The agents are also trained on full-episode trajectories, using a discount factor of 0.99.

For Multi Tasks Interleaving, the optimal hyperparameters found and used for each algoritm found and used for each algorithm can be found in 4.

|  | CCA Att | CCA LSTM | PG |
|---|---|---|---|
| Policy cost | 1 | 1 | 1 |
| Entropy Cost | 5e-2 | 5e-2 | 5e-2 |
| Forward baseline cost | 1e-2 | 5e-3 | 5e-2 |
| Conditional baseline cost | 1e-2 | 5e-3 | |
| Hindsight classifier cost | 1e-2 | 1e-2 | |
| Action independence cost | 1e1 | 1e1 | |
| Learning rate | 5e-4 | 5e-4 | 1e-3 |

**Table 4:** Multi Tasks Interleaving hyperparameters

### B.3.3 ADDITIONAL RESULTS

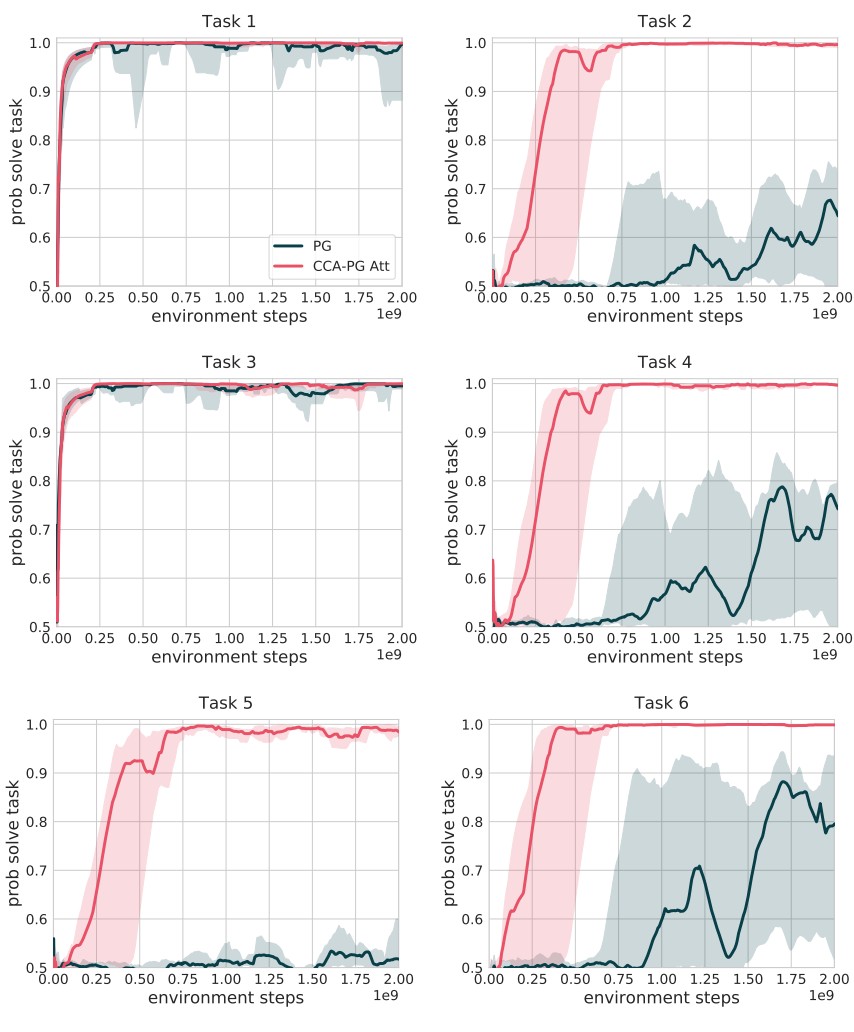

**Figure 11:** Probability of solving each task in the 6-task setup for **Multi Task Interleaving**.

As explained in 3.3, CCA is able to solve all 6 tasks quickly despite the variance induced by the exogenous factors. Actor-critic on the other hand despite solving the easy tasks 1 and 3 for which the agent receives a big reward, it is incapable of reliably solve the 4 remaining tasks for which the associated reward is smaller. This helps unpacking Fig. 5.

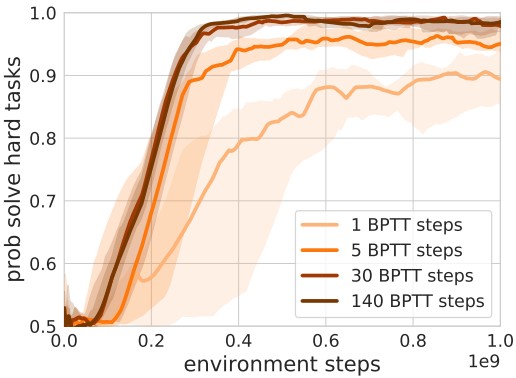

**Figure 12: Impact of the number of back-propagation through time steps performed into the hindsight function for CCA RNN.** Probability of solving the hard tasks in the 6-task setup of **Multi Task Interleaving**.

### B.3.4 ABLATION STUDY

Fig.12 shows the impact of the the number of back-propagation through time steps performed into the backward RNN of the hindsight function while performing full rollouts. This show that learning in hard tasks, where hindsight is crucial for performances, is not much impacted by the number of back-propagation steps performed into the backward RNN. This is great news as this indicates that learning in challenging credit assignment tasks can happen when the hindsight function sees the whole future but only can backprop through a limited window.

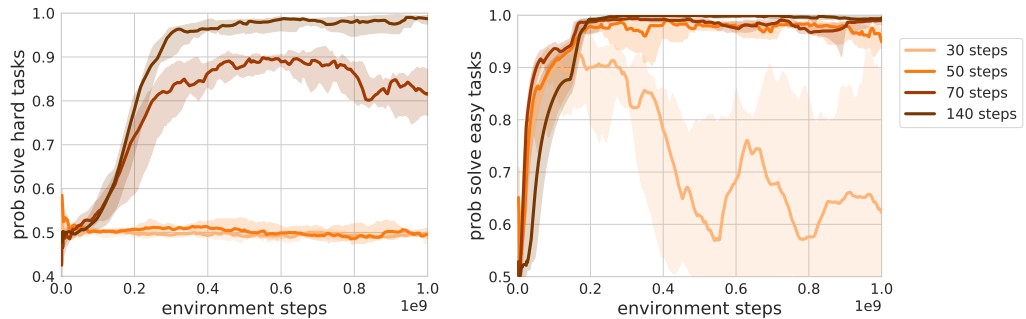

**Figure 13: Impact of the unroll length over the probability of solving hard and easy tasks for CCA RNN.** Probability of picking up the correct squares for the hard and easy tasks in the 6-task setup of **Multi Task Interleaving**.

Fig.13 shows how performances of CCA with an RNN for the hindsight function are impacted by the unroll length. As expected, the less you are able to look into the future, the harder it becomes to solve this hard credit assignment task as you become limited in your capacity to take exogenous effects into account. The two previous results are exciting because to work at its fullest CCA seems to only require to have access to as many steps into the future as possible while not needing to do back-propagation through the full sequence. This observation is really handy as the environments considered become more complex and with longer episodes.

## C  RELATION BETWEEN HCA, CCA, AND FC ESTIMATORS

The FC estimators generalizes both the HCA and CCA estimator. From FC, we can derive CCA by assuming that $\Phi_t$ and $A_t$ are conditionally independent (see next section). We can also derive state and return HCA from FC.

For return HCA, we obtain both an all-action and baseline version of return HCA by choosing $\Phi_t = G_t$. For state HCA, we first need to decompose the return into sums of rewards, and apply the policy gradient estimator to each reward separately. For a pair $(X_t, R_{t+k})$, and assuming that $R_{t+k}$ is a function of $X_{t+k}$ for simplicity, we choose $\Phi_t = X_{t+k}$. We then sum the different FC

estimators for different values of $k$ and obtain both an all-action and single-action version of state HCA.

Note however that HCA and CCA *cannot* be derived from one another. Both estimators leverage different approaches for unbiasedness, one (HCA) leveraging importance sampling, and the other (CCA) eschewing importance sampling in favor of constraint satisfaction (in the context of inference, this is similar to the difference between obtaining samples of the posterior by importance sampling versus directly parametrizing the posterior distribution).

## D PROOFS

### D.1 POLICY GRADIENTS

*Proof of Proposition 1.* By linearity of expectation, the expected return can be written as $\mathbb{E}[G] = \sum_t \gamma^t \mathbb{E}[R_t]$. Writing the expectation as an integral over trajectories, we have:

$$\mathbb{E}[R_t] = \sum_{\substack{x_0,\dots,x_t \\ a_0,\dots,a_t}} \left( \prod_{s \leq t} \left( \pi_\theta(a_s|x_s) P(x_{s+1}|x_s, a_s) \right) \right) R(x_t, a_t)$$

Taking the gradient with respect to $\theta$:

$$\nabla_\theta \mathbb{E}[R_t] = \sum_{\substack{x_0,\dots,x_t \\ a_0,\dots,a_t}} \left( \sum_{s' \leq t} \nabla_\theta \pi_\theta(a_{s'}|x_{s'}) P(x_{s'+1}|x_{s'}, a_{s'}) \left( \prod_{s \leq t, s \neq s'} \left( \pi_\theta(a_s|x_s) P(x_{s+1}|x_s, a_s) \right) \right) \right) R(x_t, a_t)$$

We then rewrite $\nabla_\theta \pi_\theta(a_{s'}|x_{s'}) = \nabla_\theta \log \pi_\theta(a_{s'}|x_{s'}) \pi_\theta(a_{s'}|x_{s'})$, and obtain

$$\nabla_\theta \mathbb{E}[R_t] = \sum_{\substack{x_0,\dots,x_t \\ a_0,\dots,a_t}} \left( \sum_{s' \leq t} \nabla_\theta \pi_\theta(a_{s'}|x_{s'}) \left( \prod_{s \leq t, s} \left( \pi_\theta(a_s|x_s) P(x_{s+1}|x_s, a_s) \right) \right) \right) R(x_t, a_t)$$

$$= \mathbb{E} \left[ \sum_{s' \leq t} \nabla_\theta \log \pi_\theta(A_{s'}|X_{s'}) R_t \right]$$

Summing over $t$, we obtain

$$\nabla_\theta \mathbb{E}[G] = \mathbb{E} \left[ \sum_{t \geq 0} \gamma^t \sum_{s' \leq t} \nabla_\theta \log \pi_\theta(A_{s'}|X_{s'}) R_t \right]$$

which can be rewritten (with a change of variables):

$$\nabla_\theta \mathbb{E}[G] = \mathbb{E} \left[ \sum_{t \geq 0} \nabla_\theta \log \pi_\theta(A_t|X_t) \sum_{t' \geq t} \gamma^{t'} R_{t'} \right]$$

$$= \mathbb{E} \left[ \sum_{t \geq 0} \gamma^t \nabla_\theta \log \pi_\theta(A_t|X_t) \sum_{t' \geq t} \gamma^{t'-t} R_{t'} \right]$$

$$= \mathbb{E} \left[ \sum_{t \geq 0} \gamma^t S_t G_t \right]$$

To complete the proof, we need to show that $\mathbb{E}[S_t V(X_t)] = 0$. By iterated expectation, $\mathbb{E}[S_t V(X_t)] = \mathbb{E}[\mathbb{E}[S_t V(X_t)|X_t]] = \mathbb{E}[V(X_t)\mathbb{E}[S_t|X_t]]$, and we have $\mathbb{E}[S_t|X_t] = \sum_a \nabla_\theta \pi_\theta(a|X_t) = \nabla_\theta(\sum_a \pi_\theta(a|X_t)) = \nabla_\theta 1 = 0$. □

*Proof of Proposition 2.* We start from the single action policy gradient $\nabla_\theta \mathbb{E}[G] =$ $\mathbb{E}\left[\sum_{t \geq 0} \gamma^t S_t G_t\right]$ and analyse the term for time t, $\mathbb{E}[S_t G_t]$.

$$
\begin{aligned}
\mathbb{E}[S_t G_t] &= \mathbb{E}[\mathbb{E}[S_t G_t | X_t, A_t]] \\
&= \mathbb{E}[S_t \mathbb{E}[G_t | X_t, A_t]] \\
&= \mathbb{E}[S_t Q(X_t, A_t)] \\
&= \mathbb{E}\left[\mathbb{E}[S_t Q(X_t, A_t) | X_t]\right] \\
&= \mathbb{E}\left[\sum_a \nabla_\theta \pi_\theta(a|X_t) Q(X_t, a)\right]
\end{aligned}
$$

The first and fourth inequality come from different applications of iterated expectations, the second from the fact $S_t$ is a constant conditional on $X_t, A_t$, and the third from the definition of $Q(X_t, A_t)$.  $\square$

## D.2   PROOF OF FC-PG THEOREM

*Proof of theorem 1.* We need to show that $\mathbb{E}\left[S_t \frac{\pi_\theta(A_t|X_t)}{\mathbb{P}_\pi(A_t|X_t, \Phi_t)} V(X_t, \Phi_t)\right] = 0$, so that $\frac{\pi_\theta(A_t|X_t)}{\mathbb{P}_\pi(A_t|X_t, \Phi_t)} V(X_t, \Phi_t)$ is a valid baseline. As previously, we proceed with the law of iterated expectations, by conditioning successively on $X_t$ then $\Phi_t$

$$
\begin{aligned}
\mathbb{E}\left[S_t \frac{\pi_\theta(A_t|X_t)}{\mathbb{P}_\pi(A_t|X_t, \Phi_t)} V(X_t, \Phi_t)\right] &= \mathbb{E}\left[\mathbb{E}\left[S_t \frac{\pi_\theta(A_t|X_t)}{\mathbb{P}_\pi(A_t|X_t, \Phi_t)} V(X_t, \Phi_t) \bigg| X_t, \Phi_t\right]\right] \\
&= \mathbb{E}\left[V(X_t, \Phi_t) \mathbb{E}\left[S_t \frac{\pi_\theta(A_t|X_t)}{\mathbb{P}_\pi(A_t|X_t, \Phi_t)} \bigg| X_t, \Phi_t\right]\right]
\end{aligned}
$$

Then we note that

$$
\mathbb{E}\left[S_t \frac{\pi_\theta(A_t|X_t)}{\mathbb{P}_\pi(A_t|X_t, \Phi_t)} \bigg| X_t, \Phi_t\right] = \sum_a \mathbb{P}_\pi(a|X_t, \Phi_t) \nabla \log \pi_\theta(a|X_t) \frac{\pi_\theta(a|X_t)}{\mathbb{P}_\pi(a|X_t, \Phi_t)} = \sum_a \nabla \pi_\theta(a|X_t) = 0.
$$

$\square$

*Proof of theorem 2.* We start from the definition of the $Q$ function:

$$
\begin{aligned}
Q(X_t, a) &= \mathbb{E}[G_t | X_t, A_t = a] = \mathbb{E}_{\Phi_t}\left[\mathbb{E}[G_t | X_t, \Phi_t, A_t = a] | X_t, A_t = a\right] \\
&= \int_\phi \mathbb{P}_\pi(\Phi = \varphi | X_t, A_t = a) Q(X_t, \Phi_t = \varphi, a)
\end{aligned}
$$

We also have

$$
\mathbb{P}_\pi(\Phi = \varphi | X_t, A_t) = \frac{\mathbb{P}_\pi(\Phi = \varphi | X_t) \mathbb{P}_\pi(A_t = a | X_t, \Phi_t = \phi)}{\mathbb{P}_\pi(A_t = a | X_t)},
$$

which combined with the above, results in:

$$
\begin{aligned}
Q(X_t, a) &= \int_\phi \mathbb{P}_\pi(\Phi = \varphi | X_t) \frac{\mathbb{P}_\pi(A_t = a | X_t, \Phi_t = \phi)}{\pi_\theta(a|X_t)} Q(X_t, \Phi_t, a) \\
&= \mathbb{E}\left[\frac{\mathbb{P}_\pi(A_t = a | X_t, \Phi_t = \phi)}{\pi_\theta(a|X_t)} Q(X_t, \Phi_t, a) \bigg| X_t\right]
\end{aligned}
$$

For the compatibility with policy gradient, we start from:

$$
\mathbb{E}[S_t G_t] = \mathbb{E}\left[\sum_a \nabla_\theta \pi_\theta(a|X_t) Q(X_t, a)\right]
$$

We replace $Q(X_t, a)$ by the expression above and obtain

$$
\begin{aligned}
\mathbb{E}[S_t G_t] =& \mathbb{E}\left[\sum_a \nabla_\theta \pi_\theta(a|X_t)\mathbb{E}\left[\frac{\mathbb{P}_\pi(A_t = a|X_t, \Phi_t = \phi)}{\pi_\theta(a|X_t)}Q(X_t, \Phi_t, a)\Big|X_t\right]\right] \\
=& \mathbb{E}\left[\mathbb{E}\left[\sum_a \nabla_\theta \pi_\theta(a|X_t)\frac{\mathbb{P}_\pi(A_t = a|X_t, \Phi_t = \phi)}{\pi_\theta(a|X_t)}Q(X_t, \Phi_t, a)\Big|X_t\right]\right] \\
=& \mathbb{E}\left[\mathbb{E}\left[\sum_a \nabla_\theta \log\pi_\theta(a|X_t)\mathbb{P}_\pi(A_t = a|X_t, \Phi_t = \phi)Q(X_t, \Phi_t, a)\Big|X_t\right]\right] \\
=& \mathbb{E}\left[\sum_a \nabla_\theta \log\pi_\theta(a|X_t)\mathbb{P}_\pi(A_t = a|X_t, \Phi_t = \phi)Q(X_t, \Phi_t, a)\right]
\end{aligned}
$$

Note that in the case of a large number of actions, the above can be estimated by

$$
\frac{\nabla_\theta \log\pi_\theta(A'_t|X_t)\mathbb{P}_\pi(A'_t|X_t, \Phi_t = \phi)}{\pi_\theta(A'_t|X_t)}Q(X_t, \Phi_t, A'_t),
$$

where $A'_t$ is an independent sample from $\pi_\theta(.|X_t)$; note in particular that $A'_t$ shall NOT be the action $A_t$ that gave rise to $\Phi_t$, which would result in a biased estimator.

### D.3  PROOF OF CCA-PG THEOREMS

Assume that $\Phi_t$ and $A_t$ are conditionally independent on $X_t$. Then, $\frac{\mathbb{P}_\pi(A_t=a|X_t, \Phi_t=\phi)}{\mathbb{P}_\pi(A_t=a|X_t)} = 1$. In particular, it is true when evaluating at the random value $A_t$. From this simple observation, both CCA-PG theorems follow from the FC-PG theorems.

To prove the lower variance of the hindsight advantage, note that

$$
\begin{aligned}
\mathbb{V}[G_t - V(X_t, \Phi)] &= \mathbb{E}[(G_t - V(X_t, \Phi_t))^2] = \mathbb{E}[G_t^2] - \mathbb{E}[V(X_t, \Phi_t)^2] \\
\mathbb{V}[G_t - V(X_t)] &= \mathbb{E}[(G_t - V(X_t))^2] = \mathbb{E}[G_t^2] - \mathbb{E}[V(X_t)^2]
\end{aligned}
$$

where the second equality comes from the fact that $\mathbb{E}[G_t V(X_t, \Phi_t)|X_t, \Phi_t] = V(X_t, \Phi_t)^2$. To prove the first statement, we have $(G_t - V(X_t, \Phi_t))^2 = G_t^2 + V(X_t, \Phi_t)^2 - 2G_t V(X_t, \Phi_t)$, and apply the law of iterated expectations to the last term:

$$
\begin{aligned}
\mathbb{E}[G_t V(X_t, \Phi_t)] =& \mathbb{E}[\mathbb{E}[G_t V(X_t, \Phi_t)|X_t, \Phi_t]] \\
=& \mathbb{E}[V(X_t, \Phi_t)\mathbb{E}[G_t|X_t, \Phi_t]] = \mathbb{E}[V(X_t, \Phi_t)^2]
\end{aligned}
$$

The proof for the second statement is identical. Finally, we note that by Jensen's inequality, we have $\mathbb{E}[V(X_t, \Phi_t)^2] \leq \mathbb{E}[V(X_t)^2]$, from which we conclude that $\mathbb{V}[G_t - V(X_t, \Phi_t)] \leq \mathbb{V}[G_t - V(X_t)]$.

$\square$

### D.4  PROOFS OF MODEL-BASED GRADIENT THEOREMS IN APPENDIX E

*Proof of Lemma 1.* The proof follows from two simple facts. The first is that the return is a deterministic function $G(X_t, a, \mathcal{E}_{t+})$. The second is that, from the law of iterated expectations we have $\mathbb{E}_{\mathcal{E}_{t+}}[G(X_t, a', \varepsilon_{t+})] = \mathbb{E}_{X_{t+}}[\mathbb{E}_{\mathcal{E}_{t+}|X_{t+}}[G(X_t, a', \varepsilon_{t+})]]$, for any distribution of $X_{t+}$. The left hand-side is $\mathbb{E}_{X_{t+} \sim p(.|X_t, a')}$. Taking the distribution of $X_{t+}$ to be $p(.|X_t, a)$, we obtain the desired result. $\square$

*Proof of Lemma 2.* The policy gradient can be written:

$$
\int_{A'_t, \mathcal{E}_{t+}} P(\mathcal{E}_{t+})\pi_\theta(A'_t|X_t)\nabla_\theta \log\pi_\theta(A'_t|X_t)G(X_t, A'_t, \mathcal{E}_{t+}) \tag{5}
$$

But we also have:

$$
P(\mathcal{E}_{t+}) = P(\mathcal{E}_{t+}|X_t) = \int_{X_{t+}, A_t} \pi_\theta(A_t|X_t)P(X_{t+}|A_t, X_t)P(\mathcal{E}_{t+}|X_{t+}, A_t)
$$

For simplicity, denote $\kappa = \pi_\theta(A_t|X_t)P(X_{t+}|A_t, X_t)P(\mathcal{E}_{t+}|X_{t+}, A_t)$. Combined with equation (5), we find:

$$\int_{X_{t+}, A_t, \mathcal{E}_{t+}, A'_t} \kappa \pi_\theta(A'_t|X_t)\nabla_\theta \log \pi_\theta(A'_t|X_t)G(X_t, A'_t, \mathcal{E}_{t+}) \tag{6}$$

Next, we analyze the same quantity but replacing $G(X_t, A'_t, \mathcal{E}_{t+})$ by $G(X_t, A_t, \mathcal{E}_{t+})$, and find:

$$\int_{X_{t+}, A_t, \mathcal{E}_{t+}, A'_t} \kappa \pi_\theta(A'_t|X_t)\nabla_\theta \log \pi_\theta(A'_t|X_t)G(X_t, A_t, \mathcal{E}_{t+}) =$$

$$\int_{X_{t+}, A_t, \mathcal{E}_{t+}} \kappa G(X_t, A_t, \mathcal{E}_{t+}) \left( \int_{A'_t} \pi_\theta(A'_t|X_t)\nabla_\theta \log \pi_\theta(A'_t|X_t) \right) = 0 \tag{7}$$

since $\left( \int_{A'_t} \pi_\theta(A'_t|X_t)\nabla_\theta \log \pi_\theta(A'_t|X_t) \right) = 0.$ $\qquad\square$

Subtracting equation (7) from (6), we obtain the desired result.

# E    RL ALGORITHMS, COMMON RANDOMNESS, STRUCTURAL CAUSAL MODELS

In this section, we provide an alternative view and intuition behind the CCA-PG algorithm by investigating credit assignment through the lens of causality theory, in particular *structural causal models* (SCMs) (Pearl, 2009a). We relate these ideas to the use of common random numbers (CRN), a standard technique in optimization with simulators (Glasserman & Yao, 1992). We start by presenting algorithms with full knowledge of the environment in the form of both a perfect model and access to the random number generator (RNG) and see how an SCM of the environment can improve credit assignment. We progressively relax assumptions until no knowledge of the environment or its random number generator is required and CCA-PG is recovered.

## E.1    STRUCTURAL CAUSAL MODEL OF THE MDP

*Structural causal models* (SCM) (Pearl, 2009a) are, informally, models where all randomness is exogenous, and where all variables of interest are modeled as deterministic functions of other variables and of the exogenous randomness. They are of particular interest in causal inference as they enable reasoning about interventions, i.e. how would the *distribution* of a variable change under external influence (such as forcing a variable to take a given value, or changing the process that defines a varaible), and about counterfactual interventions, i.e. how would a particular observed outcome (sample) of a variable have changed under external influence. Formally, a SCM is a collection of model variables $\{V \in \boldsymbol{V}\}$, exogenous random variables $\{\mathcal{E} \in \boldsymbol{\mathcal{E}}\}$, and distributions $\{p_\mathcal{E}(\varepsilon), \mathcal{E} \in \boldsymbol{\mathcal{E}}\}$, one per exogenous variable, and where the exogenous random variables are all assumed to be independent. Each variable $V$ is defined by a function $V = f_V(\text{pa}(V), \boldsymbol{\mathcal{E}})$, where $\text{pa}(V)$ is a subset of $\boldsymbol{V}$ called the parents of $V$. The model can be represented by a directed graph in which every node has an incoming edge from each of its parents. For the SCM to be valid, the induced graph has to be a directed acyclic graph (DAG), i.e. there exists a topological ordering of the variables such that for any variable $V_i$, $\text{pa}(V_i) \subset \{V_1, \ldots, V_{i-1}\}$; in the following we will assume such an ordering. This provides a simple sampling mechanism for the model, where the exogenous random variables are first sampled according to their distribution, and each node is then computed in indexing order. Note that any probabilistic model can be represented as a SCM by virtue of reparametrization Kingma & Ba (2014); Buesing et al. (2019). However, such a representation is not unique, i.e. different SCMs can induce the same distribution.

We now parameterize the MDP given in section 2.1 as a SCM. The transition from $X_t$ to $X_{t+1}$ under $A_t$ is given by the transition function $f^X$: $X_{t+1} = f^X(X_t, A_t, \mathcal{E}_t^X)$ with exogenous variable / random number $\mathcal{E}_t^X$. The policy function $f^\pi$ maps a random number $\mathcal{E}_t^\pi$, policy parameters $\theta$, and current state $X_t$ to the action $A_t = f^\pi(X_t, \mathcal{E}_t^\pi, \theta)$. Together, $f^\pi$ and $\mathcal{E}_t^\pi$ induce the policy, a distribution $\pi_\theta(A_t|X_t)$ over actions. Without loss of generality we assume that the reward is a deterministic function of the state and action: $R_t = f^R(X_t, A_t)$. $\mathcal{E}^X$ and $\mathcal{E}^\pi$ are random

variables with a fixed distribution; all changes to the policy are absorbed by changes to the deterministic function $f^\pi$. Denoting $\mathcal{E}_t = (\mathcal{E}_t^X, \mathcal{E}_t^\pi)$, note the next reward and state $(X_{t+1}, R_t)$ are deterministic functions of $X_t$ and $\mathcal{E}_t$, since we have $X_{t+1} = f^X(X_t, f^\pi(X_t, \mathcal{E}_t^\pi, \theta), \mathcal{E}_t^X)$ and similarly $R_t = R(X_t, f^\pi(X_t, \mathcal{E}_t^\pi, \theta))$. Let $X_{t+} = (X_{t'})_{t'>t}$ and similarly, $\mathcal{E}_{t+} = (\mathcal{E}_t^X, \mathcal{E}_{t'})_{t'>t}$ Through the composition of the functions $f^X$, $f^\pi$ and $R$, the return $G_t$ (under policy $\pi_\theta$) is a deterministic function (denoted $G$ for simplicity) of $X_t$, $A_t$ and $\mathcal{E}_{t+}$.

### E.2 MODEL-KNOWN POLICY GRADIENT

*In this section, we assume perfect knowledge of the transition functions, reward functions, and SCM distribution. We use the term 'model-known' rather than 'model-based' to describe this situation.*

Consider a time $t$, state $X_t$, and a possible action $a$ for $A_t$. The return $G_t$ is given by the deterministic function $G(X_t, a, \mathcal{E}_{t+})$, and the Q function $Q(X_t, a) = \mathbb{E}_{\mathcal{E}_{t+}}[G_t | X_t, A_t = a]$ is its expectation over the exogenous variables $\mathcal{E}_{t+}$. We are generally interested in evaluating the Q function difference $Q(X_t, a) - Q(X_t, a')$ for two actions $a$ and $a'$. Note in particular that the advantage can be written $A(X_t, a) = \mathbb{E}_{a' \sim \pi_\theta}[Q(X_t, a) - Q(X_t, a')])$. The Q function difference can be estimated by a difference $G(X_t, a, \mathcal{E}_{t+}) - G(X_t, a', \mathcal{E}'_{t+})$ where $\mathcal{E}_{t+}$ and $\mathcal{E}'_{t+}$ are two independent samples. If we have direct access to $\mathcal{E}_{t+}$, for instance because we we have access to a simulator *and* to its random number generator, we can use common random numbers to potentially reduce variance: $G(X_t, a, \mathcal{E}_{t+}) - G(X_t, a', \mathcal{E}_{t+})$.

Note that if actions were continuous, $G$ differentiable and $a' = a + \delta$ with $\delta$ small, the quantity becomes $\frac{\partial G}{\partial a}(X_t, a, \mathcal{E}_{t+}) \times \delta$, i.e. the gradient of the return $G$ with respect to the action $a$ (see Silver et al. 2014; Heess et al. 2015; Buesing et al. 2016). In general, we will be interested in cases where $R$ may not be differentiable. However, the example highlights that the use of gradient methods implicitly assumes the use of common random numbers, and that return differences computed with common random numbers can be seen as a numerical approximation to the gradient of the return.

Having access to the model, suppose we make a two sample estimate of the policy gradient using common random numbers, and use the return of one action as baseline for the other. The policy gradient estimate is

$$\nabla_\theta V(x_0) = \mathbb{E}_{A_t, A'_t \sim \pi_\theta, \mathcal{E}_{t+}}[S_t(G(X_t, A_t, \mathcal{E}_{t+}) - G(X_t, A'_t, \mathcal{E}_{t+}))] \tag{8}$$

where we recall that we defined $S_t$ as $\nabla_\theta \log \pi_\theta(A_t | X_t; \theta)$. In many situations this estimate will have lower variance than one obtained with a state-conditional baseline (cf. eq. (1)) since the use of common noise for $G$ will strongly correlate return and baseline (which differ only in a single argument to the function $G$).

Since $A_t$ and $A'_t$ are samples from the same distribution, the update above remains valid if we swap $A_t$ and $A'_t$; averaging the two updates, we obtain a two point policy gradient:

$$\nabla_\theta V(x_0) = \frac{1}{2}\mathbb{E}_{A_t, A'_t, \mathcal{E}_{t+}}[Y_t(G(X_t, A_t, \mathcal{E}_{t+}) - G(X_t, A'_t, \mathcal{E}_{t+}))], \tag{9}$$

where $Y_t$ denotes the score function differential $(\nabla_\theta \log \pi_\theta(A_t | X_t; \theta) - \nabla_\theta \log \pi_\theta(A'_t | X_t; \theta))$. The use of a model is required since we need returns from the same state with two different actions (note that in the case of a POMDP the same initial state would require the same history, which is often computationally excessive to do).

More generally, we could use $K$ i.i.d. samples $A_t^{(1)}, \ldots, A_t^{(K)}$ and use the leave-one-out average empirical return as a baseline for each sample, which yields

$$\nabla_\theta V(x_0) = \frac{1}{K}\mathbb{E}_{A_t^{(1)}, \ldots, A_t^{(K)}, \mathcal{E}_{t+}}\left[\sum_i \nabla_\theta \log \pi_\theta(A_t^{(i)} | X_t; \theta) \Delta_i\right], \tag{10}$$

where $\Delta_i \triangleq \left(G(X_t, A_t^{(i)}, \mathcal{E}_{t+}) - \frac{1}{K-1}\sum_{j \neq i} G(X_t, A_t^{(j)}, \mathcal{E}_{t+})\right)$.

The idea of using multiple rollouts from the same initial state to perform more accurate credit assignment for policy gradient methods has been used under the name *vine* by Schulman et al. (2015). The authors also note the need for common random numbers to reduce the variance of the multiple rollout estimate (see also Ng & Jordan 2013). Interestingly, if we replace $\Delta_i$ by the argmax of softmax of the $\Delta$, we obtain a gradient estimate similar to that of the cross-entropy method, a classical and very effective planning algorithm (De Boer et al., 2005; Langlois et al., 2019).

### E.3 Model-based Counterfactual Policy Gradient

In the previous section, we derived low-variance policy updates under the assumption that we have access to both a perfect model and its noise generation process. We will now see how model-based counterfactual reasoning allows us to address both of these restrictions, recalling results from (Buesing et al., 2019). First we briefly recall what counterfactuals are, in particular in the context of reinforcement learning. Counterfactual query intuitively correspond to question of the form 'how would this precise outcome have changed, had I changed a past action to another one?'. In a structural model that consists of outcome variables $X$ and action variables $A$ set to $a$, estimating the counterfactual outcome $X'$ under an alternative action $a'$ consists in the following three steps:

- Abduction: infer the exogenous noise variables $\mathcal{E}$ under the observation: $\mathcal{E} \sim P(\mathcal{E}|X)$.
- Intervention: Fix the value of $A$ to $a'$.
- Prediction: Evaluate the outcome $X'$ conditional on the fixed values $\mathcal{E}$ and $A = a'$.

We begin with a lemma (following results from Buesing et al. (2019)), which explains that assuming model correctness, expectations of counterfactual estimates are equal to regular interventional expectations. Denote $p(\tau|X_t)$ the distribution of trajectories starting from $X_t$ and following $\pi_\theta$, $p(\tau|X_t, a)$ the distribution of trajectories starting from $X_t$, $A_t = a$, and following $\pi_\theta$ after $A_t$, and $p(\tau|X_t, a, \mathcal{E}_{t+})$ the distribution of the trajectories starting at $X_t$, $A_t = a$, following the policy $\pi_\theta$ but forcing the value of all the SCM exogenous random variables to $\mathcal{E}_{t+}$ (note that this last distribution is in fact a deterministic quantity, since all randomness has been fixed).

**Lemma 1.** *Under the assumptions above,*

$$Q(X_t, a') = \mathbb{E}_{X_{t+} \sim p(.|X_t, a')}[G] = \mathbb{E}_{X_{t+} \sim p(.|X_t, a)} \left[ \mathbb{E}_{\mathcal{E}_{t+}|X_{t+}} \left[ \mathbb{E}_{X'_{t+} \sim p(.|X_t, a', \varepsilon_{t+})} [G] \right] \right]. \quad (11)$$

In other words, we can use SCMs to perform off-policy or counterfactual evaluation *without importance sampling*, as long as we can infer the exogenous variables of interest.

This lemma is particularly useful when using an imperfect model, which we now assume is the only model available. We denote the 'real-world' or data distribution by $p_D$ and model distributions by $p_M$. Also, let $G_D$ denote the true return function and $G_M$ the imperfect model of it. Using the model, model-based variants of equation (8) are obtained by simply replacing $p$ by $p_M$:

$$\nabla_\theta V(x_0) = \sum_t \gamma^t \mathbb{E}_{A_t, A'_t \sim \pi_\theta, \mathcal{E}_{t+} \sim p_M} [S_t(G_M(X_t, A_t, \mathcal{E}_{t+}) - G_M(X_t, A'_t, \mathcal{E}_{t+}))], \quad (12)$$

Using an imperfect model, this update could have high bias. Instead of fully trusting the synthetic data generated by the model, we can combine model data and real data in equation (11), by sampling the outer expectation with respect to $p_D$ and the inner ones with respect to $p_M$. We obtain the following counterfactual policy gradient estimate:

**Lemma 2.** *Assuming no model bias, the policy gradient update is equal to*

$$\nabla_\theta V(x_0) = \sum_t \gamma^t \mathbb{E}_{X_{t+} \sim p_D(X_t, \pi_\theta)} \left[ \mathbb{E}_{\mathcal{E}_{t+} \sim p_M(\mathcal{E}_{t+}|X_{t+}), A'_t \sim \pi_\theta} [S'_t(G'_t - G_t)] \right] \quad (13)$$

*where $S'_t = \nabla_\theta \log \pi_\theta(A'_t|X_t)$ is the score function for the counterfactual action, and where $G'_t = G_M(X_t, A'_t, \mathcal{E}_{t+})$ is the model-based counterfactual return estimate. If we explicitly marginalize out $A'_t$, we obtain:*

$$\mathbb{E}_{X_{t+} \sim p_D(X_t)} \left[ \mathbb{E}_{\mathcal{E}_{t+} \sim p_M(\mathcal{E}_{t+}|X_{t+})} \left[ \sum_a \nabla_\theta \pi_\theta(a|X_t)(G_M(X_t, a, \mathcal{E}_{t+}) - G_t) \right] \right] \quad (14)$$

Note that in contrast to eq. (13), and even when assuming a perfect model and posterior, the following update will generally be biased (we will later explain why):

$$\nabla_\theta V(x_0) \neq \sum_t \gamma^t \mathbb{E}_{X_{t+} \sim p_D(X_t)} \left[ \mathbb{E}_{\mathcal{E}_{t+} \sim p_M(\mathcal{E}_{t+}|X_{t+}), A'_t \sim \pi_\theta} [S_t(G_t - G'_t)] \right] \quad (15)$$

In equations (13) and (14) above, $X_{t+}$ is sampled by the real environment, and $\mathcal{E}_{t+}$ is from the posterior noise given the observations (which would generally be given by Bayes rule, following $P(\mathcal{E}_{t+}|X_{t+}) \propto P(\mathcal{E}_{t+})P(X_{t+}|\mathcal{E}_{t+})$). In particular, this estimate does not require access to the random number generator - instead, it 'measures' (estimates) what noise in the model *must have been* to give rise to the observations given by the real environment.

The term $G_t = G_D(X_t, A_t, \mathcal{E}_{t+})$ is the empirical real-world return, while $G'_t = G_M(X_t, A'_t, \mathcal{E}_{t+})$ is the counterfactual return that would have happened *mutatis mutandis* for action $A'_t$, under the same noise realization. A very slight modification to equation (13) is to use the environment only to sample the trajectory $X_{t+}$, but to use the model for both evaluations of the return:

$$\mathbb{E}_{X_{t+} \sim p_D(X_t, \pi_\theta)} \left[ \mathbb{E}_{\mathcal{E}_{t+} \sim p_M(\mathcal{E}_{t+}|X_{t+})} \left[ S'_t(G_M(X_t, A'_t, \mathcal{E}_{t+}) - G_M(X_t, A_t, \mathcal{E}_{t+})) \right] \right] \qquad (16)$$

This may lead to less variance, and potentially even less bias: even though $G_M$ is a biased estimate of $G_D$, some of the bias will show up in both terms and cancel out, while it would remain in $G_D - G_M$.

Note also that in the presence of model bias, it is likely that equations (13) and (16) would suffer from significantly less issues (bias and variance) than their purely model-based alternative (12), as the counterfactual updates are grounded in real data ($X_{t+} \sim p_D(X)$) and corresponding to a "reconstruction" instead of a prior sample.

### E.4 FUTURE CONDITIONAL VALUE FUNCTIONS

In the previous sub-section, we assumed knowledge of a (potentially imperfect) model but no access to the random number generation; in this sub-section, we make the inverse assumption: we assume we have access to the random number generation, but develop a model-free method that can leverage the access to the entropy engine without explicitly assuming the model.

Let us consider again the vanilla, single action policy gradient estimate:

$$\nabla_\theta V(x_0) = \mathbb{E}[S_t(G_t - V(X_t))]$$

Classically, the baseline function is assumed to be a function of $X_t$ (recall that in the POMDP case, $X_t$ includes the history of observations). If $V$ is a function of any quantity which is statistically dependent on $A_t$ conditionally on $X_t$, the baseline could result in a biased estimator for the policy gradient. A sufficient, standard assumption to guarantee this condition, is to not use any data from the future relative to time step $t$ as input for $V$, although such knowledge is available in principle in off-line policy updates. While the optimal baseline may not necessarily be a state value function, a good surrogate for determining a baseline is to minimize the variance of the advantage: for a state-dependent baseline, this corresponds to setting the baseline to the value function.

Note that in a structural causal model the random variables $\mathcal{E}$ are explicitly assumed to have a distribution affected by no other random variables, in particular $\mathcal{E}_{t+} \perp\!\!\!\perp A_t|X_t$. It is therefore valid to include them in the baseline; by the same argument as above, a strong candidate baseline is therefore $V(X_t, \mathcal{E}_{t+}) = \mathbb{E}[G_t|X_t, \mathcal{E}_{t+}]$. What does this baseline correspond to? Note that in this expectation the only randomness left is in action $a_t$; the corresponding generalized value function is in fact $V(X_t, \mathcal{E}_{t+}) = \sum_a \pi_\theta(a|s_t)G(X_t, a, \mathcal{E}_{t+})$. Learning this value function is therefore very closely related to learning the return function $G$, which itself is closely related to learning the composition of the transition and reward functions. The corresponding policy gradient becomes:

$$\mathbb{E}_{\mathcal{E}_{t+}, A_t}[S_t(G_t - V(X_t, \mathcal{E}_{t+})]] \qquad (17)$$

where $V(X_t, \mathcal{E}_{t+})$ can be learned by minimizing the square loss between a function of $X_t, \mathcal{E}_{t+}$ and empirical returns $G_t$. Note that this estimate of the advantage is also lower variance than that of $G_t - V(X_t)$, following $V(X_t) = \mathbb{E}(V(X_t, \mathcal{E}_{t+}))$ and Jensen's inequality (see Weber et al. (2019) for a proof, generalized definitions of value functions, and conditions for valid baselines; and see Weaver & Tao (2001); Greensmith et al. (2004) for results on optimal baselines for policy gradients.).

### E.5 RECOVERING MODEL-FREE COUNTERFACTUAL POLICY GRADIENTS (CCA-PG)

In the last two sections, we relaxed the assumptions of having access to either a model of the environment, or to the access to the random number generator. In this section, we combine both ideas to recover our proposed algorithm, CCA-PG.

To do so, we follow the idea from section E.4 that a future-conditional value function can be model-like and result in improved credit assignment; however, like in section E.3, instead of assuming knowledge of $\mathcal{E}_{t+}$, we will estimate it from trajectory information. Let $\mathcal{F}_t$ represent any subset or function of the trajectory, such as the sequence of states $X_{t+}$, the return, the sequence of observations, actions, etc. In MDPs, $\mathcal{F}_t$ only needs to be a function of present and future states, in POMDPs, $\mathcal{F}_t$ will need to be a function of the entire trajectory, for instance, of present and future agent state.

A first approach, related to distributional reinforcement learning (Veness et al., 2015; Bellemare et al., 2017), is to forego modeling the environment (as in section E.3) and directly model distributions over returns or value functions. We can induce such a probabilistic models over returns, by assuming a given parametrized base distribution $p_\theta(\mathcal{E})$, and approximate posterior $q(\mathcal{E}|\mathcal{F})$, and value function $V(X_t, \mathcal{E}_{t+})$. These components can be learned by the KL-regularized regression

$$\sum_t \mathbb{E}\left[\int_\mathcal{E} q(\mathcal{E}_t|\mathcal{F}_t) \log \frac{q(\mathcal{E}_t|\mathcal{F}_t)}{p(\mathcal{E}_t)} + (V(X_t, \mathcal{E}_t) - G_t)^2 + (Q(X_t, A_t, \mathcal{E}_t) - G_t)^2\right].$$

This equation intuitively captures the idea of measuring a $\mathcal{E}_t$ from $\mathcal{F}_t$ such that $\mathcal{E}_t$ is approximately independent from the trajectory (represented by the KL loss) yet good at predicting the return (represented by the value loss). We can then train a policy with counterfactual policy gradient in the following way: For each time $t$, sample $\mathcal{E}_t$ from $q(\mathcal{E}_t|\mathcal{F}_t)$, compute $V$ and $Q$ and either apply update (3) or (4).

This approach is flawed however: Even if $\mathcal{E} \perp\!\!\!\perp A_t | X_t$ is assumed to hold under the prior, this will not hold in general under the the posterior $q$, i.e. to which extent the agent *knows* about the true value of $\mathcal{E}_t$ will in general depend on $A_t$. For instance, consider a POMDP corresponding to a maze navigation task, where the only uncertainty is the maze layout. Including the maze layout in the value function will not bias the policy gradient update, and typically lower its variance. However, if we train (in a supervised fashion) a network to produce an estimate of the map given the agent's observations, and provided the value function with a hindsight estimate of the map, the resulting policy update would in general be biased. This is the same reason why equation (15) is in general biased, even assuming a perfect model and posterior.

For this reason, we forgo explicit probabilistic modeling, and choose an implicit approach, modeling a function $\Phi_t$ of the trajectory that captures information for predicting return, and therefore only implicitly perfors inference over $\mathcal{E}_t$. Following the intuition developed in this section, we require that $\Phi_t$ be independent of $A_t$ while predicting returns accurately, which finally connects back to the algorithms detailed in section 2.

## F  LINKS TO CAUSALITY AND SIMPLE EXAMPLES

In this section, we will further link the ideas developed in this report to causality theory. In particular we will connect them to two notions of causality theory known as individual treatment effect (ITE) and average treatment effect (ATE). In the previous section, we extensively leveraged the framework of structural causal models. It is however known that distinct SCMs may correspond to the same distribution; learning a model from data, we may learn a model with correct distribution but with with incorrect structural parametrization and counterfactuals. We may therefore wonder whether counterfactual-based approaches may be flawed when using such a model. We investigate this question, and analyze our algorithm in very simple settings for which closed-form computations can be worked out.

### F.1  INDIVIDUAL AND AVERAGE TREATMENT EFFECTS

Consider a simple medical example which we model with an SCM as illustrated in figure 14. We assume population of patients, each with a full medical state denoted $S$, which summarizes all factors, known or unknown, which affect a patient's future health such as genotype, phenotype etc. While $S$ is never known perfectly, some of the patient's medical history $H$ may be known, including current symptoms. On the basis of $H$, a treatment decision $T$ is taken; as is often done, for simplicity we consider $T$ to be a binary variable taking values in {1='treatment', 0='no treatment'}. Finally, health state $S$ and treatment $T$ result in a observed medical outcome $O$, a binary variable taking values in {1='cured', 0='not cured'}. For a given value $S = s$ and $T = t$, the outcome is a function

(also denoted $O$ for simplicity) $O(s,t)$. Additional medical information $F$ may be observed, e.g. further symptoms or information obtained after the treatment, from tests such as X-rays, blood tests, or autopsy.

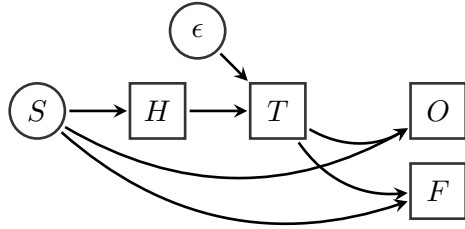

**Figure 14:** The medical treatment example as a structured causal model.

In this simple setting, we can charactertize the effectiveness of the treatment for an individual a patient with profile $S$ by the Individual Treatment Effect (ITE) which is defined as the difference between the outcome under treatment and no treatment.

**Definition 1** (Individual Treatment Effect).

$$ITE(s) = \mathbb{E}[O|S=s, \mathrm{do}(T=1)] - \mathbb{E}[O|S=s, \mathrm{do}(T=0)] = O(s, T=1) - O(s, T=0) \quad (18)$$

The conditional average treatment effect is the difference in outcome between the choice of $T=1$ and $T=0$ when averaging over all patients with the same set of symptoms $H=h$

**Definition 2** (Conditional Average Treatment Effect).

$$ATE(h) = \mathbb{E}[O|H=h, \mathrm{do}(T=1)] - \mathbb{E}[O|H=h, \mathrm{do}(T=0)] = \int_s p(S=s|H=h)(O(s,T=1) - O(s,T=0))$$
$$(19)$$

Since the exogenous noise (here, $S$) is generally not known, the ITE is typically an unknowable quantity. For a particular patient (with hidden state $S$), we will only observe the outcome under $T=0$ or $T=1$, depending on which treatment option was chosen; the counterfactual outcome will typically be unknown. Nevertheless, for a given SCM, it can be counterfactually estimated from the outcome and feedback, using the procedure detailed in section E.3 (we suppose $O$ is included in $F$ to simplify notation)

**Definition 3** (Counterfactually Estimated Individual Treatment Effect).

$$\textit{CF-ITE}[H=h, F=f, T=1] = \delta(o=1) - \int_{s'} P(S=s'|H=h, F=f, T=1)O(s', T=0)$$
$$(20)$$

$$\textit{CF-ITE}[H=h, F=f, T=0] = \int_{s'} P(S=s'|H=h, F=f, T=1)O(s', T=0) - \delta(o=1)$$
$$(21)$$

In general the counterfactually estimated ITE will not be exactly the ITE, since there may be remaining uncertainty on $s$. However, the following statements relate CF-ITE, ITE and ATE:

- If $S$ is identifiable from $O$ and $F$ with probability one, then the counterfactually-estimated ITE is equal to the ITE.
- The average (over $S$, conditional on $H$) of the ITE is equal to the ATE.
- The average (over $S$ and $F$, conditional on $H$) of CF-ITE is equal to the ATE.

Assimilating $O$ to a reward, the above illustrates that the ATE (equation 19) essentially corresponds to a difference of Q functions, the ITE (equation 18) to the return differences found in equations (8) and (17), and the counterfactual ITE to the quantities found in equations (13) and (14). In contrast, the advantage $G_t - V(H_t)$ is a difference between a return (a sample-level quantity) and a value

function (a population-level quantity, which averages over all individuals with the same medical history $H$); this discrepancy explains why the return-based advantage estimate can have very high variance.

As mentioned previously, for a given joint distribution over observations, rewards and actions, there may exist distinct SCMs that capture that distribution. Those SCMs will all have the same ATE, which measures the effectiveness of a policy on average. But they will generally have different ITE and counterfactual ITE, which, when using model-based counterfactual policy gradient estimators, will lead to different estimators. Choosing the 'wrong' SCM will lead to the wrong counterfactual, and so we may wonder if this is a cause for concern for our methods.

*We argue that in terms of learning optimal behaviors (in expectation), estimating inaccurate counterfactual is not a cause for concern.* Since all estimators have the same expectation, they would all lead to the correct estimates for the effect of switching a policy for another, and therefore, will all lead to the optimal policy given the information available to the agent. In fact, one could go further and argue that for the purpose of finding good policies in expectations, we should only care about the counterfactual for a precise patient inasmuch as it enables us to quickly and correctly taking better actions for future patients for whom the information available to make the decision ($H$) is very similar. This would encourage us to choose the SCM for which the CF-ITE has minimal variance, regardless of the value of the true counterfactual. In the next section, we elaborate on an example to highlight the difference in variance between different SCMs with the same distribution and optimal policy.

### F.2 Betting against a fair coin

We begin from a simple example, borrowed from Pearl (2009b), to show that two SCMs that induce the same interventional and observational distributions can imply different counterfactual distributions. The example consists of a game to guess the outcome of a fair coin toss. The action $A$ and state $S$ both take their values in $\{h, t\}$. Under model **I**, the outcome $O$ is 1 if $A = S$ and 0 otherwise. Under model **II**, the guess is ignored, and the outcome is simply $O = 1$ if $S = h$. For both models, the average treatment effect $E[O|A = h] - E[O|A = t]$ is 0 implying that in both models, one cannot do better than random guessing. Under model **I**, the counterfactual for having observed outcome $O = 1$ and changing the action, is always $O = 0$, and vice-versa (intuitively, changing the guess changes the outcome). Therefore, the ITE is $\pm 1$. Under model **II**, all counterfactual outcomes are equal to the observed outcomes, since the action has in fact no effect on the outcome. The ITE is always 0.

In the next section, we will next adapt the medical example into a problem in which the choice of action does affect the outcome. Using the CF-ITE as an estimator for the ATE, we will find how the choice of the SCM affects the variance of that estimator (and therefore how the choice of the SCM should affect the speed at which we can learn which is the optimal treatment decision).

### F.3 Medical example

Take the simplified medical example from figure 14, where a population of patients with the same symptoms come to the doctor, and the doctor has a potential treatment $T$ to administer. The state $S$ represents the genetic profile of the patient, which can be one of three $\{\text{GENE}_A, \text{GENE}_B, \text{GENE}_C\}$ (each with probability $1/3$). We assume that genetic testing is not available and that we do not know the value of $S$ for each patient. The doctor has to make a decision whether to administer drugs to this population or not, based on repeated experiments; in other words, they have to find out whether the average treatment effect is positive or not. We consider the two following models:

- In model **I**, patients of type $\text{GENE}_A$ always recover, patients of type $\text{GENE}_C$ never do, and patients of type $\text{GENE}_B$ recover if they get the treatment, and not otherwise; in particular, in this model, administering the drug never hurts.

- In model **II**, patients of type $\text{GENE}_A$ and $\text{GENE}_B$ recover when given the drug, but not patients of type $\text{GENE}_C$; the situation is reversed ($\text{GENE}_A$ and $\text{GENE}_B$ patients do not recover, $\text{GENE}_C$ do) when not taking the drug.

In both models - the true value of giving the drug is $2/3$, and not giving the drug $1/3$, which leads to an ATE of $1/3$. For each model, we will evaluate the variance of the CF-ITE, under one of the four possible treatment-outcome pair. The results are summarized in table 5. Under model $\mathbf{A}$, the variance of the CF-ITE estimate (which is the variance of the advantage used in CCA-PG gradient) is $1/6$, while it is 1 under model $\mathbf{B}$, which would imply $\mathbf{A}$ is a better model to leverage counterfactuals into policy decisions.

| Treatment | Outcome | Type | CF-Prob. | | CF-O | | ITE | | CF-V | | CF-ITE | | Var |
|---|---|---|---|---|---|---|---|---|---|---|---|---|---|
| Drug | Cured | GENE$_A$ | 1/2 | 1/2 | 1 | 0 | 0 | +1 | | | | | |
| | | GENE$_B$ | 1/2 | 1/2 | 0 | 0 | +1 | +1 | 1/2 | 0 | 1/2 | 1 | |
| | | GENE$_C$ | 0 | 0 | ✕ | | ✕ | | | | | | 1/6 |
| | Not cured | GENE$_A$ | 0 | 0 | ✕ | | ✕ | | | | | | 1 |
| | | GENE$_B$ | 0 | 0 | ✕ | | ✕ | | 0 | 1 | 0 | -1 | |
| | | GENE$_C$ | 1 | 1 | 0 | 1 | 0 | -1 | | | | | |
| No Drug | Cured | GENE$_A$ | 1 | 0 | 1 | 1 | 0 | 0 | | | | | |
| | | GENE$_B$ | 0 | 0 | ✕ | | ✕ | | 1 | 0 | 0 | 1 | |
| | | GENE$_C$ | 0 | 1 | 0 | 0 | 1 | 1 | | | | | 1/6 |
| | Not cured | GENE$_A$ | 0 | 1/2 | 1 | 1 | -1 | -1 | | | | | 1 |
| | | GENE$_B$ | 1/2 | 1/2 | 1 | 1 | -1 | -1 | 1/2 | 1 | -1/2 | -1 | |
| | | GENE$_C$ | 1/2 | 0 | 0 | 0 | 0 | 0 | | | | | |

**Table 5:** CCA-PG variance estimates in the medical example. CF-Probs. Red value are estimates for model $\mathbf{I}$, blue ones are for model $\mathbf{II}$. CF-Prob denotes posterior probabilities of the genetic state $S$ given the treatment $T$ and outcome $O$. CF-O is the counterfactual outcome. The ITE is the individual treatment effect (difference between outcome and counterfactual outcome). CF-V is the counterfactual value function, computed as the average of CF-O under the posterior probabilities for $S$. CF-ITE is the counterfactual advantage estimate (difference between O and CF-V). Var is the variance of CF-ITE under the prior probabilities for the outcome.

