# OpenReview forum: "Model-Free Counterfactual Credit Assignment"
_ICLR.cc/2021/Conference — Reject_

### Official Review · AnonReviewer2 · 2020-10-28
**Interesting but confusing**

**Rating:** 5
**Confidence:** 4

**Review:**


The message and paper propose a couple of environments where there is exogenous noise added to the reward function and the particular method in the paper specifically looks at this type of noise. While the method proposed may work in these types of environments it's not clear if more interesting environments do have these properties and we should be more concerned with this problem or that the environments used in the paper were specifically constructed to fit the use case of the algorithm.

The proposed method in the paper does offer interesting insight into how certain temporary consistent variables and the identification of such variables can help decrease the variance over policy estimates. However, the results in the paper are not overly convincing with respect to understanding the importance of this method on more realistic tasks that the community is generally interested in.

Some more detailed notes:
- The introduction does not state that the particular credit assignment problems being looked into is that of partially observed environments. Overall, I find the writing in the introduction to not motivate the problem well our lead the reader towards what to expect in the rest of the paper. This makes it very difficult to understand and appreciate the paper.
- If it's still not clear from the middle section to let the detail of the contribution is going to be period by this point it sounds like the method is just going to be a modification to a q function.
- There does not appear to be my significant information on how the mutual information metric is computed between the action space and latent variable space.

---

> ### Author Response · Authors · 2020-11-20
> **Response**
>
> We firmly believe that whether we study reinforcement learning as a model for human cognition, or as a toolbox for solving complex problems, our problems are in some aspects more realistic than classical RL environments. Classical RL environments exhibit few of the credit assignment issues associated with real world problems: most benchmarks are deterministic or nearly so; and the agent is in a very controlled environment where its action directly affects the outcome; no externalities or exogenous affect the outcome; different tasks are clearly separated and not interlaced; and in most setups, the agent is the sole actor in the environment. None of these assumptions are verified in reality.
>
> Detailed notes:
>
> *The introduction does not state that the particular credit assignment problems being looked into is that of partially observed environments*
> Our approach is not limited to partially observed environments.
>
> *it sounds like the method is just going to be a modification to a q function.*
> We are not sure what this sentence means. What does ‘a modification to a q function’ mean, and how would it invalidate a method?
>
> *There does not appear to be my significant information on how the mutual information metric is computed between the action space and latent variable space.*
> We will include these details in the main text.

---

> > ### Comment · AnonReviewer2 · 2020-11-21
> > **Environments and more**
> >
> > While I agree with the comments on how many current RL benchmarks are lacking this does not convince me that the environments used in the paper are much better. If this can be addressed in the updated version of the paper this can help the readers understand the more general benefit of the work.
> >
> > " a modification to a q function"
> > As R1 has noted the method adds Φt to the Q function Q(Xt , Φt , a) that can be learned. It is not clear how this is learned. It would help to convincingly outline how this modification is important and as some of the other reviewers have noted compare to prior methods.

---

> > > ### Author Response · Authors · 2020-11-23
> > > **Environments**
> > >
> > > We chose to keep the environments relatively simple visually and in terms of control in order to tease out the credit assignment aspects. Combining all aspects is at this point too difficult to solve in our mind (their combinations would constitute interesting environments, but they prove too challenging for typical RL setups at the moment).
> > >
> > > If you believe no environments without complex visuals can be interesting, we will only have to agree to disagree. Otherwise, we would welcome you to elaborate on why, precisely, you find these environments not interesting? It is difficult to address the criticism or come up with alternative environments if we are not provided with more details.
> > >
> > > Regarding the training of Phi: All the details for training Phi can be found in the appendix 1. We will shortly submit a version with those details (slightly abridged) in the main text. Note that ‘modifications to baseline and q-functions’ are very common research topics in methodological RL and approximate inference papers. But the modification is by no means trivial. Consider a general learning rule for providing signal at all actions (‘counterfactual learning’, so to speak):
> > > \sum_a \grad \pi(a|x) S(a), where S(a) is the learning signal for action a. To our knowledge, very few RL papers offer any generally valid rule beyond using the vanilla Q function S(a)=Q(s,a).
> > >
> > > Conditioning on any arbitrary information Q(x,a,phi) will generally not work. We identify the criterion to make these updates be correct. We are not aware of other RL work that provide alternative forms of the learning signal for all actions, with the exception of HCA, which provides a single alternative rule (instead of a family of rules), with no clear guarantees about its performance.

---

> > > > ### Comment · AnonReviewer2 · 2020-11-24
> > > > **Comments on revisions and Environments**
> > > >
> > > > It is true that using simpler, more pure environments help gauge if the method is performing as intended. However, to indicate that the method is of larger interest to the community it is beneficial to display that it improves performance on common environments in the community or even environments that are currently difficult.
> > > >
> > > >
> > > > Section 2.6 provides many important details in understanding how the method works and is actually trained. The terminology of "we assume the agent..." makes it difficult for the reader to clearly understand what the agent is truly doing or what exactly has been implemented. However, with a careful read, it can be understood.

---

> > > > > ### Author Response · Authors · 2020-11-24
> > > > > **comment**
> > > > >
> > > > > Regarding your first paragraph, this is a fair point. While common RL benchmarks may have limited credit assignment issues, we could modify a common RL environment to exacerbate those issues (such as assigning all rewards to the final state, or adding high variance perturbations to the dynamics).
> > > > >
> > > > > For your second paragraph, would “We additionally assume that at training time, a hindsight network processes the entire trajectory to compute hindsight statistics Phi […]. These statistics are then used to compute the hindsight value function V_\theta(X_t,Phi_t)“. If not, what is the source of confusion exactly? Does the diagram in the appendix help?

---

### Official Review · AnonReviewer4 · 2020-10-28
**Novel and interesting work on hindsight counterfactuals with perhaps some missing evaluations**

**Rating:** 5
**Confidence:** 3

**Review:**

This work attempts to address the problem posed by high reward variance and low sample efficiency in model-free RL algorithms. The proposal is to use counterfactuals to do finer-grained credit-assignment and reasoning about alternative actions without having to learn a potentially difficult environment model.

This is done by conditioning the value function on a random variable $\phi$ that attempts to capture everything else about the future trajectory not resulting from the current action. This is done by maximizing the independence between $\phi$ and $A$ given the current state. A classifier that predicts action based on $\phi$ is required to do the above. This is also learned from data.

Claimed contributions:
Proposing a set of environments with difficult credit assignment.
Novel algorithms that use counterfactuals that are unbiased and guarantee lower variance.

+ The approach seems novel and interesting.
+ The claimed contributions are supported to a large extent by theory and experimentation.
+ The idea of constructing value functions conditioned on future trajectory information is not novel (Hindsight Credit Assignment does this), but the idea of learning the conditioning variable is (HCA uses states or returns).
+ The paper is clearly written. The illustrative example of counterfactuals in hindsight with Alice and Megan is helpful.
+ The approach is evaluated first on a bandit task and then on different versions of a partially observable gridworld environment and finally on a multi-task setting.
+ Comparison to vanilla policy gradient and a couple of versions of prior work (HCA) over a substantial number of random seeds.
+ The task interleaving setting is an interesting benchmark for multi-task settings.

This work builds off of HCA and mainly addresses the case of high variance in rewards where the prior work seems to fail. It performs similar to vanilla PG on environments with little randomness in reward for similar actions, but better than HCA.

The authors claim that they do not require a model of the environment but a classifier $h(A_T|X_T, \phi)$ is learned which resembles an inverse model. Even though the approach does not require building a forward model, I am curious to know the performance of a model-based approach such as by Buesing et al. trained on the same data available for $h(A_T|X_T, \phi)$ in these environments. Is it difficult to learn a model for the proposed tasks?

I think the work contains enough novelty, the writing is clear and the experimentation is extensive. But, I am unsure whether to recommend acceptance without a model-based baseline trained on data available to the classifier used in this approach.

---

> ### Author Response · Authors · 2020-11-20
> **Response**
>
> We thank you for your encouraging review.
>
> It is true that while model-free, our approach attempts at capturing aspects of model-based reasoning. However, a classifier is a far simpler object to learn than a full model of an environment.
>
> A counterfactual model-based approach as in Buesing et al. would probably solve the problem. `Classical’ model-based approaches may be more difficult to tune because they will also be affected by the problem variance, and therefore may result in inaccurate models.
>
> We believe however that generally speaking, model-based approaches are still significantly more complicated to set in motion. Model-based RL is still a developing field, and there are far more design choices involved in designing a model-based RL architecture than a model-free one. Choices have to be made with regards to the RL algorithm itself, the environment model, how is data used to learn the agent and the world model, the losses used. We are afraid that an attempt at a model-based approach would result in an unfair comparison, in which the model-based approach would underperform, which is not what we aim to prove. We aim to show instead we can get the benefits of a model-based approach without some of the drawbacks. Nevertheless, we are happy to take suggestions as to what a ‘fair’ model-based comparison would be. Would you want it to be counterfactual as well, or more classical?

---

### Official Review · AnonReviewer3 · 2020-10-28
**Official Blind Review #3**

**Rating:** 6
**Confidence:** 3

**Review:**

#### Summary:
The paper explores a new approach to credit assignment that complements existing work. It focuses on model-free approaches to credit assignment using hindsight information. In contrast to some prior work on this topic, e.g., (Harutyunyan et al. 2019), the paper does not rely explicitly on hand-crafted information, but instead learns to extract useful hindsight information. The contributions of the paper are two-fold. First, the paper introduces two new policy gradient estimators, FC-PG and CCA-PG, and it proves that the novel gradient estimators are unbiased. Second, it provides experimental evidence that the novel estimators are beneficial compared to some prior work (in particular (Harutyunyan et al. 2019)).


#### Comments:

Overall, I found the contributions of the paper interesting, but I'm somewhat on the fence about this paper due to the following pros and cons.

Pros: The paper is clearly written, easy to follow, and interesting to read. It provides a good overview of the related work, and motivates well the problem at hand. Furthermore, the paper showcases that its algorithmic approach has theoretical grounding, and it experimentally verifies that it's beneficial compared to concurrent approach from (Harutyunyan et al. 2019).

Cons: Given that a very similar type of counterfactual credit assignment approach has already been proposed in prior work, the technical contributions (theorems) of the paper seem somewhat incremental. The experiments, while indicating potential benefits of the proposed approach, utilize relatively simple environments compared to some of the recent papers on credit assignment (e.g. (Arjona-Medina et al. 2019), (Guez et al 2019)). Moreover, the experiments could include more state of the art baselines.

Apart from these high level comments, the following comments include suggestions for improvements and questions.

Related work: Since the hindsight credit assignment of (Harutyunyan et al. 2019) is a special case of FC-PG, this connection should be mentioned earlier in the paper, not just in the related work section. The flow of the paper is currently misleading, given that there is prior work that does propose quite similar ideas, e.g., the content between the title to section 2.4 does not seem to be reflect relevant prior work. Perhaps referencing relevant papers in earlier sections, or moving the related work section, would resolve this issue.

Notation: Notation in the paper often omits important dependences, making some of the calculations confusing or not immediately clear. In the interest of making the claims more precise, it would be very useful to add important dependencies where needed. For example, in equation (1), does $P(a|X_t, \Phi_t)$ depend on policy $\pi$? Moreover, the notation does not seem to be consistent, e.g., policy $\pi$ sometimes has dependency on $\theta$ sometime not (in gradient calculations).

Appendix: I think adding some parts from the appendix could improve the clarity of the content. In particular, the last paragraph on Page 3 that starts with 'We ensure that these statistics...' is not providing sufficient explanations regarding the technical content important for understanding the results. It is also not clear if all the content in the appendix is relevant for the results described in the main text.

Minor typos:
-removed from the advantage, resulting a significantly lower variance estimator. --- resulting in?
-$\lambda_{IM}$ does not seem to be defined before being used (in the paragraph before section 3.2)
-and the the benefits of the more general FC-PG and all-actions estimators. --- remove one 'the'?

#### Questions:

A) I'm a bit puzzled by the discussion regarding the conditional independence requirement in Section 2.5. Why is this an 'intuitive' requirement? How does it influence the interpretation in the  paragraph before Theorem 3? How does this compare to  (Harutyunyan et al. 2019) argument that '$h(.)$ quantifies the relevance of action a to the future state $X_k$'?

B) The proof of Theorem 3 and Theorem 4 in Section D3 says that the theorems follows from Theorem 1 and Theorem 2 given the conditional independence assumption. Could you explain in more detail why the second statement (about variance) in Theorem 3 follows from Theorem 1 and 2?

C) How does this approach compare to Ferret et al.: Self-Attentional Credit Assignment for Transfer in Reinforcement Learning?

---

> ### Author Response · Authors · 2020-11-20
> **Response**
>
> We thank you for your thoughtful review and comments.
>
> We challenge the statement that the work is incremental. The CCA estimator is novel and does not have similar ideas in the literature we are familiar with. CCA was developed concurrently with HCA; their main (and intriguing) similarity was requiring learning a hindsight classifier in both cases. As a result, we spent a significant amount of time trying to understand the connections, and came up with the FC estimator, (which does resemble HCA). However, as pointed in the appendix, you cannot derive CCA from HCA and vice-versa, they are fundamentally different estimators leveraging different ideas (similar to the difference between variational inference and sampling-based methods in inference).
>
> Our current proof for CCA is derived from FC, but it is possible to prove CCA directly without invoking the h/pi ratio, which makes the connection less clear. We believe that presenting the unified approach clarifies the connection but should not be used as an argument that CCA and HCA are the same; they are not. Note further that CCA provides a performance guarantee (in terms of lower variance) and a guiding principle in terms of deriving useful Phi.  We will mention HCA earlier in the paper (while trying not to have the discussion in two parts of the paper).
>
> For state of the art baselines, we believe our ideas are orthogonal to many ideas in state of the art RL algorithms. CCA could be combined with natural policy updates (MPO, V-MPO), off-policy learning, better representation learning, and so on. We worry that these comparisons may therefore bring confounding factors and we are not convinced of their value. As an example, is it meaningful to compare CCA (vanilla policy gradient + counterfactual credit assignment) and  VMPO (natural policy gradient + vanilla credit assignment). If we ran e.g. VMPO on our tasks and found it to underperform, we would not want the reader to conclude VMPO is worse than CCA. Their benefits are likely complimentary. Nevertheless, we are happy to try to include additional baselines, would there be any in particular you are interested in seeing the results of?
>
> Note that Guez et al. is not a paper on credit assignment; it does representation learning. Arjona-Medina deals with a slightly different setup (delayed reward, though they do lead to increased variance).
>
> Notation wise:You are right, P(a|X_t,\Phi_t) is implicitly a function of pi. It would be better to make that dependency explicit, so we will add it in the paper. We will also fix the notation for pi through the paper, thanks for noticing.
> Re: appendix, see general response- we will move elements of the appendix to the main text.
>
> Typos: Thank you, will fix.
>
> Questions:
>
> A) Perhaps the following is helpful: generally, we can think of a trajectory as a function of two factors: the agent’s actions, and external factors, which are independent of the action. The external factors are not known to the agent, but some of them may have a strong effect on the outcome. Phi represents the agents’ attempt at measuring those external factors from trajectory. Those external factors are defined both by having affected the outcome (i.e. predictive of the return), and being exogenous, i.e. not caused by the agent (hence the independence assumption). By ‘removing’ the contribution of those external factors to the outcome, all that is left is the agent’s actions (skills).
>
> Regarding the comment *h(.) quantifies the relevance of action a to the future state Xk*, note that state-HCA is a special case of the all-action FC PG estimator, which is different from CCA. As discussed in the paper, it is harder to find a good criterion for Phi in the all action case (we however still offer some leads). Note however that the intuition about removing action information still holds. Suppose for instance that the agent state Xk includes all past actions (A1...Ak). In this case it is easy to show that the HCA estimator degenerates into the vanilla, single action, policy gradient estimator, as carrying too much information about actions in the hindsight statistics makes the agent incapable of understanding more precise counterfactuals. We can elaborate on that proof if you would like.
>
> B) Good catch, we forgot to include that bit. We will add it back.
>
> C) This work mainly focuses on the problem of credit assignment for transfer in RL which is not directly related to the points we are making in this paper. However, we would be happy to include it.  Ferret et. al leverage transformers to derive a heuristic to perform reward shaping. While we also investigate the use of transformers, our approach is not based on explicit reward shaping.

---

### Official Review · AnonReviewer1 · 2020-10-29
**Unclear and vague, but can be improved. I could not find how the authors find the key component of the method, i.e., phi.**

**Rating:** 3
**Confidence:** 3

**Review:**

In this paper, the authors develop a new policy gradient method to reduce the variance in the gradient estimations.
In the commonly used policy method, the bias is a function of the state. e.g., V(x_t). In this paper, the authors propose to use bias V(x_t,\phi_t) where \phi_t is a statistics of future events such that \phi_t is conditionally independent of the action at time t.

The authors show that using such statistics in V(x_t,\phi_t) results in a reduction in the gradient estimate used in policy gradient methods.


Later, the authors also show that their method performs well in practice.


There is a set of problems with the paper's presentation, which resulted in the negative evaluation.
The analysis in the paper is straightforward and also easy to follow. However, I could not find how the proposed algorithm learns the \phi.

I encourage the authors to improve the clarity, presentation, and language in this paper.

1) I did not get what the authors mean by luck or skill. These terms do not seem to be coherent terms in this paper. I highly encourage the authors to rethink such usage. Unless the authors mathematically define it in the paper.

2) "Another issue of model-free methods is that counterfactual reasoning, i.e. reasoning about what would have happened had different actions been taken with everything else remaining the same, is not possible."

Can the authors clarify it? Why is it not? When I learn a Q function, that tells me what would be the expected return if I choose other actions following the same policy, right?
If you mean evaluating other policies is not possible, I still doubt the statement is true.

3) "Given a trajectory, model-free methods can in fact only learn about the actions that were actually taken to produce the data, and this limits the ability of the agent to learn quickly."

Can you clarify this? I can use function approximation based methods, and then, the first part of the authors' statement is no longer true. The second statement is inaccurate since the author did not quantify with respect to what method the quickness in learning is compared to.

4) "actions taken by the agent will only affect a vanishing part of the outcome". What do the authors mean here? What the vanishing part of the outcome refers to?

5) "mak- ing it increasingly difficult to learn from classical reinforcement learning algorithms", what the authors mean by learning from classical RL algorithm? and why the authors think a better credit assessment is needed and is the way to go. What motivates the authors to state the issue is the credit assignment?

6) "Second, removing the value function V (Xt) from the return Gt does not bias the estimator and typically reduces variance". Would the author refer to a paper stating that removing the value function V (Xt) from the return Gt typically reduces variance?

7)"This estimator updates the policy through the score term; note however the learning signal only updates the policy πθ(a|Xt) at the value taken by action At = a "
I am not sure I understand this sentence. Is πθ(a|Xt) the policy, or it is πθ. Do authors have a different model for each state and action pair? Even in that case, since the need to normalize action probability, changing πθ(a|Xt) will affect other πθ(a|X) as well. Therefore, I am not sure what the authors mean here.

8) Distinction between single action and all actions.
In both propositions 1 and 2, it seems that the learning signal is provided for both actions. It is not clear to me how the authors make the distinction. Especially here
"The policy gradient theorem from (Sutton et al., 2000), which we will also call all-action policy gradient, shows it is possible to provide learning signal to all actions,".
I am not sure what the authors mean.

The authors state that"A particularity of the all-actions policy gradient estimator is that the term at time t for updating the policy ∇π(a|Xt)(Q(Xt, a) depends only on past information;" but it seems to me that Q is a function of the measure on the future. Isnt it the case?

9) To motivate the usage of phi, the authors talk about a scenario in a soccer game, which again I could not find useful, especially when they bring luck and skill.
The authors state that "When using the single-action policy gradient estimate, the outcome of the game being a victory, and ,assuming a ±1 reward scheme, all her actions are made more likely".
How is it possible that all actions become more likely? when their probabilities should be sum to one?
I am not sure again. Are the authors talking about using one trajectory for all the estimates? The update in proposition 1 shows that in the case the agent action does not change the outcome, then the gradient is zero.


10) The authors state that
"In contrast, if the agent could measure a quantity Φt which has a high impact on the return but is not correlated to the agent action At , it could be far easier to learn Q(Xt , Φt , a)."
It is not clear why learning Q(Xt, a) is harder than Q(Xt , Φt , a). So far, Q(Xt, a) seems an easier function to approximate and most likely needs a fewer sample to learn Q(x, a) than something presumably complicated like Q(x, \phi , a).

11) In section 3.1, I strongly encourage the authors to elaborate more clearly on what they do. Is W a scaler? if yes, then how F can be constructed?

Do you draw U,V,W each time step??



12) Aside from many unclear statements in this paper that the authors can easily address, I could not find how the authors find \phi. Since this is the main key component of the paper, it would be great if the authors could explain it in depth. I also could not find it clear in the appendix.

13) I strongly encourage the authors to expand their study on plain MDP before getting to the POMDP complication. It is not clear where the performance gain comes from.


................................................................

Post rebuttal. The confidence rating is reduced.
I might have been mistaken, but the authors might find this paper useful. "Troubling Trends in Machine Learning Scholarship"
Again, I might be wrong, and the mentioned paper might be of no use here.

---

> ### Author Response · Authors · 2020-11-19
> **Response [1/4]**
>
>
> Thank you for your review and questions. We answer your questions below, and will ensure the updated revision reflects those clarifications.
>
> 1] *I did not get what the authors mean by luck or skill. These terms do not seem to be coherent terms in this paper. I highly encourage the authors to rethink such usage. Unless the authors mathematically define it in the paper.*
>
> The ‘luck vs skill’ metaphor is only here to guide intuition (though our method goes beyond disentangling a simple notion of luck). In RL learning via policy gradients, agents reinforce actions that led to outcomes with higher reward than expected. Those higher rewards could have been obtained through a skillful choice of action, or because of ‘luck’ ( ie exogenous variables not under the control of the agent). When both factors affect the outcome, it can be hard to understand what is the contribution of the choice of action and of external factors. Say for example a person starts a business and gets very successful. Did they get lucky, having essentially bet on the right horse (the business is in an area that would get much higher demand than expected), or did they have great intuition?
>
> *2] "Another issue of model-free methods is that counterfactual reasoning, i.e. reasoning about what would have happened had different actions been taken with everything else remaining the same, is not possible." Can the authors clarify it? Why is it not? When I learn a Q function, that tells me what would be the expected return if I choose other actions following the same policy, right? If you mean evaluating other policies is not possible, I still doubt the statement is true.*
>
> This is an interesting question; RL practitioners typically argue that Q functions provide a counterfactual, as they provide an average estimate of the reward for other actions. We argue this is a very limited counterfactual (they are technically *not* counterfactual in the sense of causality theory per Pearl), because they average over all possible outcomes with a similar starting state. What we mean by counterfactual is a finer notion: 'what would have happened in this very same episode (which is what we mean by ‘everything else remaining the same’), had I taken another action?'.
>
> To explain the difference, let us consider a very simple example. At the start of the day you receive a weather report (state x) that tells you there is a 50/50 percent chance of rain. You have to decide whether to take an umbrella or not (action a).
> If it rains and you carry an umbrella, you get a reward of 1, but if you don’t, you get a reward of -1 for getting soaked. Conversely, if it does not rain and you have an umbrella, you get a reward of -1 (due to umbrellas being cumbersome to carry around for no reason), and +1 if you don’t carry an umbrella.
>
> In this scenario, the Q(x,a) function, where x={the weather report} and a={carrying an umbrella or not} is 0 for both actions. This is because in the system described above, carrying an umbrella or not is reward-equivalent.
> Now imagine that you decide not to carry an umbrella, and get rained on (R=-1). A ‘true’ counterfactual here corresponds to understanding that *on that particular day* (in this particular episode), carrying an umbrella would have in fact resulted in a reward of +1 (and no 0 as the vanilla Q function indicates).
>
> Note this intuition can be formalized using our CCA estimator. In this example, an agent could discover that Phi=’presence of rain’ affects the rewards, but is not caused by carrying an umbrella or not (though a superstitious agent would probably believe it does).
> Q(report, no umbrella, rain) is the factual outcome (evaluate to -1), while Q(report, umbrella, rain) is the episode-specific counterfactual one, which would evaluate to +1.
>
> *4] "actions taken by the agent will only affect a vanishing part of the outcome". What do the authors mean here? What the vanishing part of the outcome refers to?*
>
> By vanishing we mean decreasing to the point of becoming very small. If you consider realistic environments in which an agent is a small part of a large system (due to the presence of complex, hard to model stochasticity as well as many other agents), the agent will typically only affect a small part of the overall trajectory of the system.

---

> ### Author Response · Authors · 2020-11-20
> **Response [2/4]**
>
> *5] "making it increasingly difficult to learn from classical reinforcement learning algorithms", what the authors mean by learning from classical RL algorithm? and why the authors think a better credit assessment is needed and is the way to go. What motivates the authors to state the issue is the credit assignment?*
>
> We mean the vast majority of model-free RL algorithms (policy gradient, Q learning and all their variants) that do not perform credit assignment beyond temporal one. The issue with credit assignment is precisely the one mentioned above: if an agent does not understand the fine grained effect of its actions and takes credit for all changes in the world, it cannot understand efficiently how to act. There will be too many confounding variables on all their actions, which make it essentially impossible to actually learn to act in the world.
>
> *3] "Given a trajectory, model-free methods can in fact only learn about the actions that were actually taken to produce the data, and this limits the ability of the agent to learn quickly." Can you clarify this? I can use function approximation based methods, and then, the first part of the authors' statement is no longer true..*
>
> *7]"This estimator updates the policy through the score term; note however the learning signal only updates the policy πθ(a|Xt) at the value taken by action At = a " I am not sure I understand this sentence. Is πθ(a|Xt) the policy, or it is πθ. Do authors have a different model for each state and action pair? Even in that case, since the need to normalize action probability, changing πθ(a|Xt) will affect other πθ(a|X) as well. Therefore, I am not sure what the authors mean here.*
>
> *8] Distinction between single action and all actions. In both propositions 1 and 2, it seems that the learning signal is provided for both actions. It is not clear to me how the authors make the distinction. Especially here "The policy gradient theorem... shows it is possible to provide learning signal to all actions,". I am not sure what the authors mean.*
>
> *9] To motivate the usage of phi, the authors talk about a scenario in a soccer game, which again I could not find useful, especially when they bring luck and skill. The authors state that "When using the single-action policy gradient estimate, the outcome of the game being a victory, and ,assuming a ±1 reward scheme, all her actions are made more likely". How is it possible that all actions become more likely? when their probabilities should be sum to one? I am not sure again. Are the authors talking about using one trajectory for all the estimates? The update in proposition 1 shows that in the case the agent action does not change the outcome, then the gradient is zero.*
>
> All these misunderstandings are related, so we address them together. We do use function approximation (in the form of neural networks), and it’s true that over time, with a large number of trajectories, agents can learn to interpolate and understand how actions are related to one another.
>
> While this is valuable, this is orthogonal to the fact that the learning signal itself may provide information about only the action which was taken (the random variable At), or other counterfactuals actions (a for other values than At).
>
> By using the chain rule, for a softmax policy parametrized by a neural networks, the gradient of the loss with respect to parameters is the gradient of the loss with respect to the set of action logits times the gradient (jacobian) of the action logits with respect to the weights. The first term, defined by the choice of the RL algorithm, is the one we mean by ‘learning signal’. The second term is a function of the particular neural network architecture. While we use deep RL, our contribution is with respect to an RL algorithm. The benefits of the neural network architecture are orthogonal to that.
>
> *‘Even in that case, since the need to normalize action probability, changing πθ(a|Xt) will affect other πθ(a|X) as well.’*
>
> This is technically correct, but the learning signal obtained by normalization is trivial and uninformative when the number of actions is large.  As an example, consider learning a classifier through supervised learning. At every step, the agent is shown an image, outputs an action in the form of a label, and is only told whether they got the answer right or wrong, then moves to the next image with no additional feedback. It’s true that getting the answer wrong slightly increases the probability of all other labels, but this is barely learning at all (and indeed we don’t recommend learning a classifier that way).
>
> But one could envisage getting feedback beyond the reward, perhaps not the label per se, but a hint (‘this is a mammal’), in which case even though you wrongly guessed dog, you know now to reinforce the probabilities of all mammals, and also decrease the probabilities of other classes than the one that was wrongly guessed.
>
> (continued)

---

> ### Author Response · Authors · 2020-11-20
> **Response [3/4]**
>
> *I am not sure what the authors mean.*
>
> The all action gives an informative learning signal (in the form of Q(x,a) ) for all actions a, not just the action At that was used in that particular trajectory.
>
> *How is it possible that all actions become more likely? when their probabilities should be sum to one*
>
> The possessive (her actions) simply meant to refer to the collections of all sampled actions through time, i.e. (A1,A2….). This is not referring to the set of all actions for a fixed time step. All the actions that Alice actually took through the game are made more likely through the gradient step \sum_t \grad \log P(At|Xt).
>
> *The update in proposition 1 shows that in the case the agent action does not change the outcome, then the gradient is zero.*
>
> We are not sure what you are referring to here. The agent may not know they have not affected the outcome if they believe they always affect the entire outcome, which is, again, the working assumption of model-free RL. In the example given, the gradient is certainly not zero.
>
>
> *6] "Second, removing the value function V (Xt) from the return Gt does not bias the estimator and typically reduces variance". Would the author refer to a paper stating that removing the value function V (Xt) from the return Gt typically reduces variance?*
>
> This is a classical RL result, which you can find in pretty much all policy gradient papers, we suggest [1-5].
> Here’s some intuition; the estimator is St(Gt-V). Its expectation is unaffected by the choice of V, so the variance is driven entirely by E[St^2 (G_t-V)^2]. In a tabular setting the score function is upper bounded by 1, which leads to an upper bound of E[(G_t-V)^2], which justifies learning the value function by minimizing the expected square advantage, and in particular will outperform the choice V=0. When function approximation is involved, for smooth functions the variance is still upper bounded by a constant time E[(G_t-V)^2].
>
> *The authors state that"A particularity of the all-actions policy gradient estimator is that the term at time t for updating the policy ∇π(a|Xt)(Q(Xt, a) depends only on past information;" but it seems to me that Q is a function of the measure on the future. Isnt it the case?*
>
> No, Q(Xt,a) is a prediction of the future. The inputs to the Q-function are computed entirely from past information (observations up to time t). Q-functions are trained on predicting future return, but the learned function does not require any input information from times t’>t to be computed (unlike, say, the return).
>
> *10] The authors state that "In contrast, if the agent could measure a quantity Φt which has a high impact on the return but is not correlated to the agent action At , it could be far easier to learn Q(Xt , Φt , a)." It is not clear why learning Q(Xt, a) is harder than Q(Xt , Φt , a). So far, Q(Xt, a) seems an easier function to approximate and most likely needs a fewer sample to learn Q(x, a) than something presumably complicated like Q(x, \phi , a).*
>
> This is a subtle point. First note that both functions approximate the return, and one has access to strictly more information (Φt), so in practice, your point is not true - it’s easy for the agent to ignore Φt if it’s not informative. In theory, the difficulty of learning the average (through monte-carlo return) is driven by its variance. The variance of the target of Q(Xt,a) is Var(Gt|Xt,a), which is higher on average than the variance Var(Gt|Xt,Φt,a) of the target of Q(Xt,Φt,a). This is because of the law of total expectation: Var(Gt|Xt,a) = E[Var(Gt|Xt,Φt,a)]  + Var[E[Gt|Xt,Φt,a]]. The second term is non-negative, hence the inequality. Let us give a simple example (similar to our bandit problem).
>
> Assume that Gt = K + N(a,1), where K is a gaussian random variable with mean 0 and large standard deviation.
> Q(a)=a, but to learn it, we are using samples with variances K^2+1. However, if you are given K in hindsight, the variance of the targets of a linear regression Q(K,a) = K+a only have variance 1, which is easy to learn.
>
> *11] In section 3.1, I strongly encourage the authors to elaborate more clearly on what they do. Is W a scaler? if yes, then how F can be constructed? Do you draw U,V,W each time step??*
>
> Thank you, this is a typo, W is in R^K. The variables U,V,W are constant across all episodes. You can think of it as a random MDP. U,V,W is sampled separately for each seed, but otherwise kept constant across times.
>
> *12] Aside from many unclear statements in this paper that the authors can easily address, I could not find how the authors find \phi. Since this is the main key component of the paper, it would be great if the authors could explain it in depth. I also could not find it clear in the appendix.*
>
> This is all detailed in appendix A1 and A2. We will bring the most important elements in the main text.

---

> ### Author Response · Authors · 2020-11-20
> **Response [4/4]**
>
>
> *13] I strongly encourage the authors to expand their study on plain MDP before getting to the POMDP complication. It is not clear where the performance gain comes from.*
>
> The performance gain comes from the lower variance estimator with no additional bias. The lower variance comes from the fact that the value function and critics leverage additional information that correlate to the return, and therefore have lower error in predicting that return. The absence of additional bias comes from the fact that the Phi are independent from the agent’s action, a fact supported by theory and practice (see figure 1 right). There is no fundamental difference between POMDPs and stochastic MDPs beside the fact that the state should be the concatenation of past observation; some of the environments we studied are essentially MDPs (the partial observability of the key to door environment makes the environment a bit more challenging, but most of the difficulty comes from the variance of rewards).
>
> We humbly believe your confidence score may have been stated too high.
>
> [1] Simple Statistical Gradient-Following Algorithms for Connectionist Reinforcement Learning, Williams
>
> [2] Likelihood Ratio Gradient Estimation for stochastic systems, Glynn
>
> [3] Monte Carlo Methods in Financial Engineering, Glasserman
>
> [4] Variance Reduction Techniques for Gradient Estimates in Reinforcement Learning, Greensmith et al.
>
> [5] Policy Gradient Methods for Reinforcement Learning with Function Approximation, Sutton et. al

---

> ### Author Response · Authors · 2020-11-23
> **Reply**
>
> Do you have additional questions / clarifications needed, or is the paper clearer at this point?

---

### Author Response · Authors · 2020-11-20
**General response**

We thank the reviewers for their thoughtful comments on our work.
Most reviewers agreed our paper presented a theoretically grounded, novel algorithm with strong performance compared to baselines that include recent related work.

Next week, we will upload an updated version with the following changes:
* More implementation details regarding the independence maximization loss will be included in the main text.
* Clarify some of the writing and fix typos.

Some concerns were raised regarding baselines and environments; we commented on these to the relevant reviewers. If we agree on which experiments would make the most sense to include, we would add them in the final version of the paper, but unfortunately cannot commit to including these in next week’s version, due to the time required to run these experiments.

---

### Author Response · Authors · 2020-11-24
**Revision**

Dear reviewers,

Thank your for your feedback which we have incorporated in a revised version of the paper. Based on your suggestions, we have :
- clarified the text and notations in several places.
- cited relevant references earlier in the paper and in the literature section.
- brought significant implementation details of the CCA-PG algorithm from the appendix into the main text (please read sections 2.5 and 2.6 in particular to find the new material).
- Added back the proof for reduced variance in the appendix.

---

### Decision · Program_Chairs · 2021-01-07
**Final Decision**

**Decision:**

Reject

**Comment:**

In this paper, the authors aim to develop a new method for credit assignment, where certain types of future information is conditioned on.  The authors are well-aware that naive conditioning on future information introduces bias due to Berkson's paradox (explaining away), and introduce a number of corrections (described in section 2.4 and 2.5).

The authors illustrate their approach via a number of simulation studies and constructed problems.

I think it would be nice if the authors found a way of connecting their notion of counterfactual to one used in causal inference (for instance, I think there is a connection via e.g. importance correction terms).

Reviewers were worried about the contribution being incremental given existing work (from 2019), and relative simplicity of the evaluation of the approach, compared to existing similar work.